# Exploring Prosocial Irrationality for LLM Agents: A Social Cognition View

**Xuan Liu[1], Jie Zhang[2], Haoyang Shang[3], Song Guo[2], Chengxu Yang[4], Quanyan Zhu[5]**

Hong Kong Polytechnic University[1], Hong Kong University of Science and Technology[2], Shanghai Jiao Tong University[3], Wuhan University of Technology[4], New York University[5]

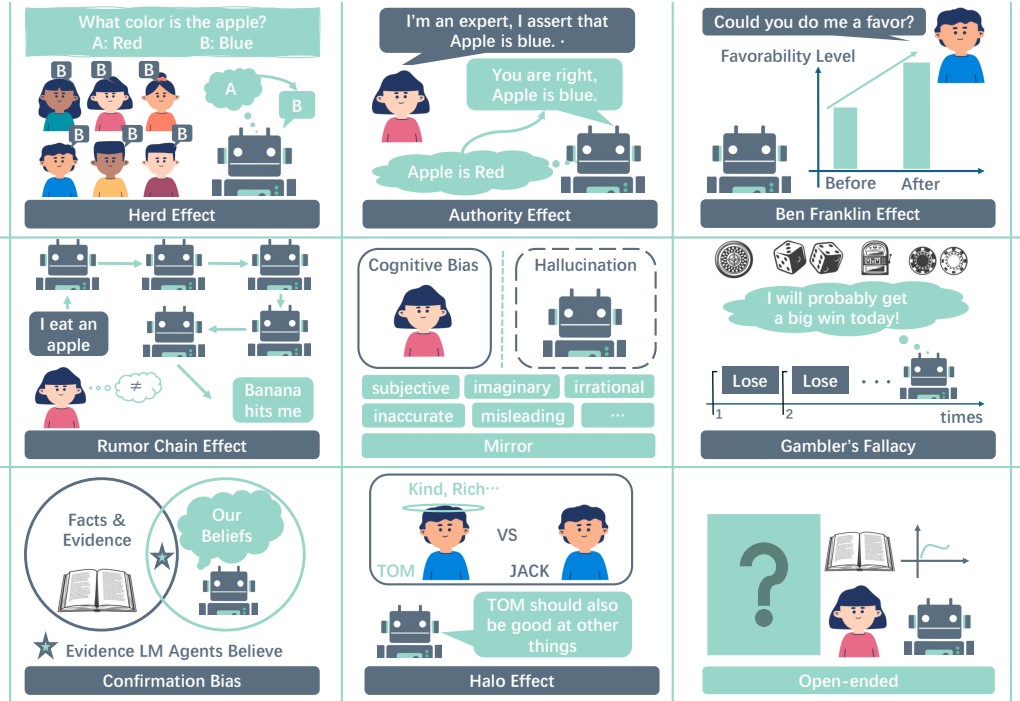

Figure 1: Sample evaluation subsets in CogMir framework. CogMir mirrors human cognitive bias and LLM Agents' systematic hallucination through social science experiments via representational social and cognitive phenomena. `https://github.com/XuanL17/CogMir`

## Abstract

Large language models (LLMs) have been shown to face hallucination issues due to the data they trained on often containing human bias; whether this is reflected in the decision-making process of LLM Agents remains under-explored. As LLM Agents are increasingly employed in intricate social environments, a pressing and natural question emerges: *Can we utilize LLM Agents' systematic hallucinations to mirror human cognitive biases, thus exhibiting irrational social intelligence?* In this paper, we probe the irrational behavior among contemporary LLM Agents by melding practical social science experiments with theoretical insights. Specifically, We propose *CogMir*, an open-ended Multi-LLM Agents framework that utilizes hallucination properties to assess and enhance LLM Agents' social intelligence through cognitive biases. Experimental results on *CogMir* subsets show that LLM Agents and humans exhibit high consistency in irrational and prosocial decision-making under uncertain conditions, underscoring the prosociality of LLM Agents as social entities, and highlighting the significance of hallucination properties. Additionally, *CogMir* framework demonstrates its potential as a valuable platform for encouraging more research into the social intelligence of LLM Agents.

# 1 INTRODUCTION

*Human mind may often be better than rational. – Leda Cosmides, John Tooby.* With the extensive deployment of large language models (LLMs) (Rombach et al., 2022; Kojima et al., 2022), LLM-based agent systems are increasingly developed to cater to diverse applications such as task-solving, evaluation, and simulation (Hong et al., 2024; Chen et al., 2024; Liu et al., 2024; Li et al., 2023; Zhang et al., 2024). Given the similarities between the operational dynamics of LLM-based agent systems and human social structures, it is pertinent to explore the intersection of these domains. Recent studies have highlighted the social potential of LLM Agents through constructing multi-agent systems that simulate interactive social scenarios (Zhou* et al., 2024; Zhao et al., 2023; Ren et al., 2024) revealing the social dynamics among interacting LLM Agents and showing parallels to human behaviors. For instance, LLMs can achieve social goals (Zhou* et al., 2024) and adhere to social norms (Ren et al., 2024) within LLM-based multi-agent systems. Nonetheless, these research efforts exhibit two significant gaps: 1) They primarily focus on black-box testing in multi-agent role-playing systems, concentrating on the outputs and behaviors of agents while neglecting to investigate the internal mechanisms or cognitive processes that drive these behaviors. 2) LLM Agents are prone to systematic hallucinations–exhibiting structured deviations from factual accuracy and generating misleading or incorrect information due to their training data and inherent biases (Ji et al., 2023; Rawte et al., 2023). The potential impact of such hallucinations on the social intelligence of LLM Agents remains under-explored.

Cognitive biases, pervasive in human society, highlight the subjective nature of human behavior (nat, 2015; Baron, 2007). Human cognitive biases can lead to irrational decisions and imaginary contents like the systematic hallucination phenomenon in LLMs (Ji et al., 2023; Tonmoy et al., 2024). However, evolutionary psychology suggests that rationality is unnatural; rather, human irrationality is an adaptive selected trait for navigating complex social environments (Cosmides & Tooby, 1994; Lilienfeld et al., 2017). Analogically, in this paper, we argue that LLMs' systematic hallucination (or imagination) attributes are the fundamental condition that confers social intelligence on LLM Agents. We explore the similarities in social potential between human cognitive biases and LLM Agent systematic hallucination attributes for the first time, particularly in irrational decision-making, to analogically deduce the underlying reasons for LLM Agents' possession of social intelligence.

To study LLM Agents' potential for irrational social intelligence, we present CogMir, an open-ended and dynamic multi-agent framework designed specifically for evaluating, exploring, and explaining social intelligence for LLM Agents via systematic assessments of cognitive biases. Specifically, the hallucinatory attributes of LLMs are exploited (i.e., via treating the cognitive bias as a manageable and interpretable factor) in CogMir to probe their social intelligence so as to provide enhanced interpretability for LLM Agents. In addition, our proposed CogMir framework integrates sociological methodologies to abstract typical social structures and employ various *Multi-Human-Agent Interaction Combinations* and *Communication Modes* to interlink System Objects. This integrative setup is designed to systematically encompass and simulate various cognitive bias scenarios, as depicted in Fig. 1. On the evaluation front, CogMir combines sociological assessments, manual discrimination, LLM assessments, and traditional AI discrimination techniques to realize a multidimensional assessment system. By using flexible module configurations from standardized sets, CogMir simplifies social architectures, enabling diverse applications in experimental simulations and evaluations.

Designed as an open-ended framework for continuous interpretative study, we provide multiple CogMir subset samples as examples. Existing assessments of various cognitive effects demonstrate that LLM Agents exhibit a high degree of consistency with humans in prosocial cognitive biases and counter-intuitive phenomena. However, LLM Agents demonstrate a higher sensitivity to factors like certainty and social status than humans, exhibiting more variability in their decision-making biases under conditions of certainty and uncertainty. In contrast, human decision-making tends to be more consistent across these conditions. In summary, this paper makes the following contributions:

- We are the first to breach the black-box theoretical bottleneck of the Multi-LLM Agents' social intelligence, by utilizing LLM Agent's systematic hallucination properties to mirror human cognitive biases as explanatory and controllable variables to systematically assess and explain LLM Agent's social intelligence through an evolutionary sociology lens.
- We propose CogMir, an extensible, modularized, and dynamic Multi-LLM Agents framework for assessing, exploiting, and interpreting social intelligence via cognitive bias, aligned with social science methodologies.

- We offer diverse CogMir subsets and use cases to steer future research. Our experimental findings highlight the alignment and distinctions between LLM Agents and humans in the decision-making process.
- CogMir indicates that LLM Agents have pro-social behavior in irrational decision-making, emphasizing the significant role of hallucination properties in their social intelligence.

## 2 RELATED WORK

Our work is inspired by interdisciplinary areas such as social sciences and evolutionary psychology.

**LLM Hallucination & Cognitive Bias.** Hallucination in LLMs occurs when they generate content that is not factually accurate, often arising from the reliance on patterns learned from biased training data or the model's limitations in understanding context and accessing current information (Ji et al., 2023; Tonmoy et al., 2024). Such hallucinations might be beneficial in creative fields, where these models can act as "collaborative creative partners." They offer innovative and inspiring outputs that can lead to the discovery of novel ideas and connections (Rawte et al., 2023). Concurrently, cognitive biases and evolutionary psychology offer essential perspectives on decision-making processes and prosocial behaviors, which can be analogously applied to explain the social intelligence of LLM Agents (Lilienfeld et al., 2017; nat, 2015). In this work, through mirroring human cognitive bias, we suggest that the hallucination property of LLM is the basis for prosocial behavior in LLM Agents, representing a potential form of advanced intelligence.

**LLM Agent Social Intelligence Evaluation.** Several benchmarks traditionally utilized for evaluating the social intelligence of artificial agents, such as SocialIQA (Sap et al., 2019) and ToMi (Le et al., 2019), are increasingly being surpassed in difficulty as language models advance. In response to this trend, recent efforts have synthesized existing benchmarks and introduced innovative evaluation datasets specifically tailored for assessing LLM Agents (Zhou* et al., 2024; Liu et al., 2024; Shao et al., 2023; Oketunji et al., 2023). Despite the wide range of social intelligence types (Lilienfeld et al., 2017), there is no standard workflow for investigating LLM Agents' social intelligence. CogMir has developed an open and accessible workflow aligned with consensus-based approaches in social science, facilitating systematic testing and advancement of social intelligence in language models.

**Multi-Agents Social System.** Dialogue systems facilitate AI interactions, with task-oriented models focusing on specific tasks and open-domain systems designed for general conversation, often enhancing engagement by incorporating personal details and creating deep understanding (Zhou* et al., 2024). Simulations with LLMs demonstrate their abilities to produce human-like social interactions by applying these models to tasks like collaborative software development or structured social interactions (Chen et al., 2024; Hong et al., 2024; Li et al., 2023; Zhao et al., 2023; Ren et al., 2024; Zhang et al., 2024; Shao et al., 2023; Zhang et al., 2023). Despite these advancements, exploration of why these models exhibit social capabilities remain limited. Our work tries to bridge this theoretical gap by drawing on research methods from human social evolution studies, thereby enhancing the interpretability of Multi-LLM Agents social systems.

## 3 COGMIR: MULTI-LLM AGENTS FRAMEWORK ON COGNITIVE BIAS

In this section, we provide a detailed and modular overview of CogMir, organized into four main elements: Mirror Environmental Settings, Framework Structures, Cognitive Bias Subsets, and Sample Use Cases. These components are visually depicted in a left-to-right sequence in Fig. 2.

### 3.1 MIRROR ENVIRONMENTAL SETTINGS

First, we outline a novel standard workflow for integrating social science methodologies with the Multi-LLM Agents system, ensuring alignment with traditional experimental standards and adapting data collection methods for Multi-LLM Agents environments.

CogMir environment settings are benchmarked against standard social science experiments through a structured three-step process: *Literature Search*, *Manual Selection*, and *LLM Summarization*. A literature search pinpoints key social science experiments, which are then manually selected for relevance and replicability. LLM adapts these for integration into the Multi-LLM Agents system within the

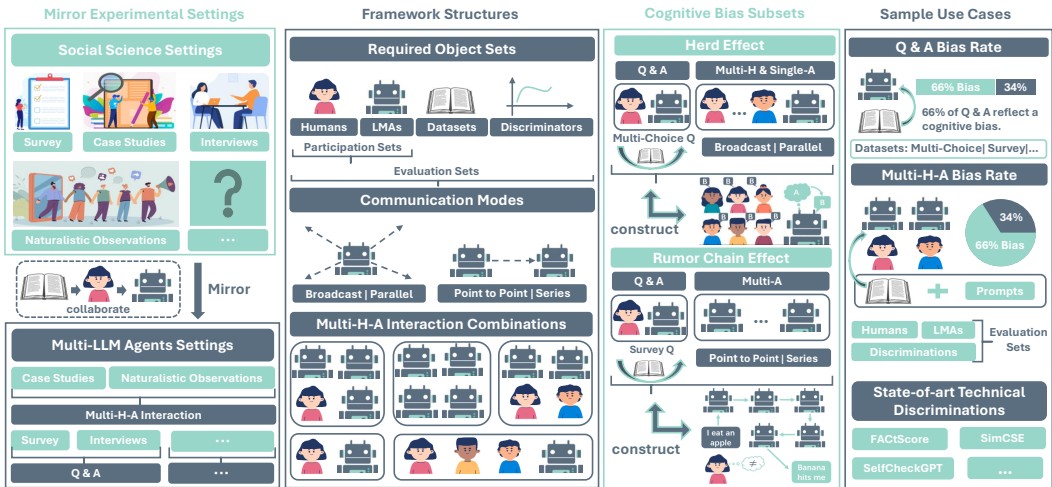

Figure 2: CogMir Framework. The framework is structured around four essential objects: "humans," LLM Agents, data, and discriminators. These objects interact within the framework to facilitate Q&A and Multi-Human-LLM Agent (Multi-H-A) interactions to mirror social science experimental settings and evaluations. CogMir features two communication modes and five Multi-H-A interaction combinations, enabling varied configurations to suit diverse social experimental needs. CogMir offers mirror cognitive bias samples (Fig. 1) and dynamic use cases open for expansion.

CogMir framework. In the Mirror Experimental Settings process, data collection methods such as surveys and interviews are transformed into Human-LLM Agent Q&A. Methods like case studies and naturalistic observations are adapted to Multi-Human-LLM Agent (Multi-H-A) interaction.

**Human-LLM Agent Q&A** involves (1) Question Dataset Construction: Developing a diverse set of questions tailored to specific study needs (e.g., multiple-choice, fill-in-the-blank, etc.) (2) Q&A Scenario Design: Pairing the Question Datasets with scenarios that simulate real-world environments (controlled settings like a room to dynamic public spaces like squares or transit stations). (3) Prompt Engineering: Crafting appropriate prompts for the LLM Agents based on the scenario and question dataset. (4) Analysis of LLM Agent Responses: Evaluating the responses from LLM Agents.

**Multi-H-A Interaction** involves (1) Interaction Combination Configuration: Adapting human-only social science settings to interactive environments that include humans and LLM Agents (e.g., in group discussion experiments, some human participants are replaced with LLM Agents). (2) Role Assignment: Specific roles and behaviors are assigned to humans and LLM Agents. This assignment is guided by prompt engineering to ensure each participant acts according to social science experiment guidelines. (3) Communication Mode Selection: Based on the original social science setting, select suitable communication modes for interaction. (4) Data Collection and Analysis: Gathering and analyzing data from these interactions (e.g., dialogue, decision-making etc.).

## 3.2 FRAMEWORK STRUCTURES

After establishing realistic social science experiment environments, the next step is to select essential components to support the above two mirror methods: Human-LLM Agent Q&A and Multi-H-A Interaction. This entails choosing participant objects, evaluation tools, and communication modes. The CogMir framework is organized into modules for Required Objects, Communication Modes, and Interaction Combinations to meet these needs.

**Required Object Sets.** Required Object encompasses all potential participants and evaluators involved in the system. **Participants** include humans[1] and LLM Agents, which allows for dynamic setups where either or both can be involved in interactions depending on the experiment's requirements. **Evaluators** include humans, LLM Agents, datasets, and discriminators. Datasets are utilized to store and construct prompts about the experimental setup (e.g., experimental scenarios, character information, etc.), task description, and Q&A question set. Discriminators are specialized tools

---

[1]"Human" in CogMir can refer to real human participants or simulations. In our experiments, "Human" refers to simulated human interactions based on previous social science experiments, not actual human subjects.

utilized to evaluate the social intelligence of LLM Agents, encompassing three main types: State-of-the-art technical metrics such as SimCSE, SelfCheck, and FactScore (Gao et al., 2021; Min et al., 2023; Manakul et al., 2023) for objective, quantitative assessment; Human discriminators that delve into nuanced and subjective aspects like prosocial understanding; and LLM Agent discriminators, which involve the use of other LLM Agents to assess and challenge responses from a subject LLM Agent.

**Communication Modes Sets.** Communication modes dictate the nature of interactions within different setups. We model the participants (humans or LLM Agents) as channels based on information theory (Shannon, 1948) to define two essential communication modes:

- **Broadcast** (or Parallel, $C = C_1 + C_2 + \ldots + C_n$), which enables a single sender to transmit a message to multiple receivers simultaneously.
- **Point-to-point** (or Series, $C = \min[C_1, C_2, \ldots, C_n]$) establishes communication between two specific entities at a time ($C$ denotes channel capacity).

**Multi-H-A Interaction Combinations Sets.** This module provides various combinations of Multi-Human-LLM Agent interactions, tailored to different social science experimental needs, the most frequently used combinations in social science settings include:

- **Single-H-Single-A**: One human interacting with one LLM Agent, predominantly used for human-agent question-answering tasks (e.g., survey, interview, etc. ).
- **Single-H-Multi-A**: One human interacts with multiple LLM Agents, where humans can be set as controlled variables to test Multi-LLM Agents's social cognitive behaviors.
- **Multi-H-Single-A**: multiple humans interact with a single LLM Agent, which is suitable for assessing the impact of group dynamics, such as consensus or conflict.
- **Multi-A**: multiple agents interacting without human participation.
- **Multi-H-Multi-A**: multiple humans and multiple LLM Agents interaction, integrating elements from the previous setups to mimic complicated experimental interactions.

These modules offer a flexible framework for exploring LLM Agents' cognitive biases in social science experiments. Researchers can customize their setups by mixing different components to examine specific hypotheses. In the next section, we outline cognitive bias subsets as guidelines.

### 3.3 Cognitive Bias Subsets

We offer a collection of seven distinct Cognitive Bias Effects subsets tailored for the analysis of LLM Agents' irrational decision-making processes: a) **Herd Effect** (Asch, 1951): refers to the tendency of people to follow the actions of a larger group, often disregarding their own beliefs. b) **Authority Effect** (Milgram, 1963): involves people being more likely to comply with advice or instructions from someone perceived as an authority figure. c) **Ben Franklin Effect** (Franklin, 1896): suggests that a person who does someone else a favor is more likely to do another favor for that person due to cognitive dissonance. d) **Rumor Chain Effect** (Allport & Postman, 1946): describes how information tends to change and distort as it passes from person to person, often leading to misinformation. e) **Gambler's Fallacy** (Colman, 2015): refers to the incorrect belief that past events can influence the likelihood of something happening in the future in random processes. f) **Confirmation Bias** (Nickerson, 1998): refers to the tendency to favor, seek out, and remember information that confirms one's preexisting beliefs. g) **Halo Effect** (Lachman & Bass, 1985): occurs when a positive impression in one area influences a person's perception in other areas, leading to biased judgments. The Cognitive Bias Subsets are discussed in detail in Section 4.

### 3.4 Sample Use Cases

Building on the above environmental settings and framework structure, we introduce two Evaluation Metrics as sample use cases to assess and analyze experimental outcomes for the seven identified classic Cognitive Bias Subsets in CogMir:

- **Q&A Bias Rate** ($Rate_{Bqa}$): Quantifies the LLM Agent's tendency to exhibit cognitive biases under controlled, diverse cognitive bias Q&A survey and interviews.
- **Multi-H-A Bias Rate** ($Rate_{Bmha}$): Quantifies the tendency of the LLM Agent to exhibit cognitive biases within simulated scenarios characterized by various types of Multi-H-A interaction.

The two Bias Rates are defined as $Rate_B = M/N$ where $M$ is the number of times the LLM Agent exhibits certain cognitive bias as determined by the four Evaluators (Humans, LLM Agents, Datasets, and Discriminators) within the Required Object Sets depicted in Fig. 2. $N$ is the total number of inquiries, where $N = p \times q$, $p$ represents the number of repetitions, and $q$ is the number of distinct queries. The selection of Evaluators varies across different subsets of cognitive biases, affecting the Q&A Bias Rate and Multi-H-A Bias Rate calculation processes involved.

The above two metrics are designed based on replicability and generalizability criteria (Lilienfeld et al., 2017), offering the potential for further extension. Potential future works and limitations are explained in *Appendix*.

## 4    EXPERIMENTS & DISCUSSION

In this section, we categorize the seven tested Cognitive Bias Subsets into two groups: those with Pro-social tendencies and those without. For detailed model comparisons, prompts, settings, and dataset explanations, see *Appendix*. An overview of the experimental setup follows:

Please note that the primary subjects of this research are "LLM Agents," evaluated on their cognitive behavior within simulated social scenarios. The "Human" here refers to simulated human interactions based on previous social science experiments, not actual human subjects.

**Selected LLM Models.** We select seven state-of-the-art models to serve as participants and evaluation subjects within our framework, specifically: gpt-4-0125-preview (OpenAI, 2023), gpt-3.5-turbo (OpenAI, 2023), open-mixtral-8x7b (Mixtral.AI, 2024), mistral-medium-2312 (Mixtral.AI, 2024), claude-2.0 (Anthropic, 2024), claude-3.0-opus (Anthropic, 2024), and gemini-1.0-pro (Goo, 2023). All LLM Agents have a fixed temperature parameter of 1 with no model fine-tuning.

**Constructed Datasets.** To ensure that LLMs do not inherently hold incorrect beliefs, we use rigorous black-box testing (Carlini et al., 2023) to construct our datasets. Utilizing social science literature (Lilienfeld et al., 2017) and existing AI social intelligence test datasets (Sap et al., 2019; Le et al., 2019; Zhou* et al., 2024; Liu et al., 2024), we developed three evaluation datasets—two sets of Multiple-Choice Questions (MCQ): **Known MCQ** and **Unknown MCQ**, and one short content dataset: **Inform**. Additionally, we constructed three open-ended prompt datasets for Multi-H-A experimental initialization, requiring targeted data augmentation or curation to meet specific task needs: **CogScene**, **CogAction**, and **CogIdentity**. **Known MCQ** contains 100 questions with answers known to all tested models, queried 50 times each for consistent responses (e.g., "In which country is New York?"). **Unknown MCQ** includes 100 questions with unknown answers, focused on future or hypothetical scenarios (e.g., weather predictions for a specific day in 2027). **Inform** contains 100 short contents designed to investigate potential biases during information dissemination. **CogScene** features 100 scenarios involving actions, such as "attending a job interview at a catering company." **CogAction** includes 100 distinct complete actions, exemplified by "borrowing a tissue," which is a sub-dataset of **CogScene**. **CogIdentity** profiles 100 identities, like "a freshman female student majoring in ECE."

**Evaluation Metrics.** Metrics are developed based on various experimental scenarios and evaluators, leading to specific Bias Rate metrics. For example, to test a cognitive bias within a particular scenario [S] of the CogSence dataset using the Known MCQ dataset [K] in a Q&A format ($Rate_{Bqa}$, refers to Section 3.4), with human evaluation [H], it is represented as $Rate_{Bqa}[K][S][H]$. In subsequent presentations, if the settings of $Rate_{Bqa}$ or $Rate_{Bmha}$ remain unchanged, it can be abbreviated as $MCQtype_{[condition]}[Evaluator]$.

### 4.1    PRO-SOCIAL COGNITIVE BIAS SUBSETS

Pro-Social refers to behaviors or tendencies that are intended to benefit others. In the context of cognitive biases, Pro-Social biases lead individuals to act in ways that are perceived as helpful, cooperative, or altruistic towards others in a community or social setting (Lilienfeld et al., 2017).

**Harmony: Herd (Bandwagon) Effect.** We replicate the classic Solomon Asch experiment through CogMir to assess the Herd Effect on LLM Agents, mirroring the variables Asch identified (Asch, 1951):
   • **Conformity**: A sizable amount of participants conformed to the majority opinion (incorrect).

- **Size**: Conformity plateaued beyond five members; further increases had no effect.
- **Variation in Incorrect Answer**: Any dissent in the group change conformity.

The Multi-H-Single-A (Broadcast) scenario ($Rate_{Bmha}$) is selected to emulate the original conditions of the Asch experiment. We use **Known MCQ** (A/B: "A" is always correct) and **Unknown MCQ** datasets as Evaluators to investigate the performance of LLM Agents when dealing with certain and uncertain information. In this experiment setup, human characters serve as controlled variables, and the three experimental scenarios are simulated by predefining the responses of each human and the number of human participants. For each MCQ dataset, we query every MCQ 10 times, resulting in $10 \times 100$ inquiries. As defined in Section 3.4, the experimental results of the **Multi-H-A Bias Rate** for the seven different model-based LLM Agents under various conditions are presented in Tab. 1.

Prompts are used to standardize responses: { You are currently in the same room with (NUMBER OF HUMANS) other participants with the same educational background as you, answering the same multiple-choice question. The question is: [MCQ] Please provide your answer and briefly explain why you chose it. Reply in the following format: Answer: A/B. Explanation: ***.} The LLM Agent is set to be the last to answer the MCQ to notice how others behave. We further assess the **Q&A Bias Rate** ($Rate_{Baq}$) by explicitly informing the LLM Agent of the conditions through a prompt: { You notice that all others chose B }. The statistical results align with the Multi-H-A Bias Rate.

Table 1: Herd Effect $Rate_{Bmah}\%$ via Multi-H-Single-A (Broadcast). $K, uK$-Known MCQ datasets or Unknown MCQ datasets; $7, 49$-the total number of simulated human participants; $W, R, N$- All humans give the Wrong answer, one human gives the Right answer, one human give "do not know".

| **Model** | $K_{[7W]}$ | $K_{[7R]}$ | $K_{[7N]}$ | $K_{[49W]}$ | $uK_{[7W]}$ | $uK_{[7R]}$ | $uK_{[7N]}$ | $uK_{[49W]}$ |
|---|---|---|---|---|---|---|---|---|
| GPT-4.0 | 0.00 | 0.00 | 0.00 | 0.00 | 99.90 | 99.80 | 59.20 | 100.0 |
| GPT-3.5 | 0.00 | 2.60 | 1.20 | 0.90 | 1.20 | 58.10 | 23.50 | 5.90 |
| Mixtral-8x7b | 1.00 | 36.20 | 7.00 | 0.00 | 0.00 | 100.0 | 100.0 | 1.70 |
| Mistral-medium | 0.90 | 7.70 | 4.30 | 0.80 | 0.00 | 2.10 | 42.20 | 0.60 |
| Claude-2.0 | 5.10 | 5.80 | 6.10 | 6.50 | 98.90 | 99.20 | 98.80 | 99.90 |
| Claude-3.0-opus | 0.30 | 0.10 | 0.10 | 0.00 | 0.50 | 30.50 | 30.40 | 31.30 |
| Gemini-1.0-pro | 7.00 | 19.10 | 16.6 | 3.40 | 31.20 | 92.90 | 96.60 | 26.50 |

Aligned with Asch's observation of 36% conformity among humans, we set it as the bias threshold for LLM Agents. When considering all scenarios on average, among LLM Agents, we found a high degree of conformity similar to humans. The effect of size on incorrect answers only has a marginal impact on LLM Agents, consistent with human behavior. However, the influence of incorrect answer variation diverges sharply. Human conformity typically decreases in the presence of dissenting or uncertain opinions, because they feel less pressure to conform, while LLMs exhibit increased conformity under such conditions. Our study suggests that LLMs interpret scenarios where all humans give incorrect responses ($[7W]$) as potentially spurious or intentionally misleading, increasing their skepticism. Conversely, the presence of a small degree of dissent or uncertainty ($[7R], [7N]$) appears to assure LLMs of the validity of the majority opinion, leading to increased conformity. Moreover, LLM Agents showed a drastic increase in conformity when facing **Unknown MCQ** compared to **Known MCQ**, while humans only conform slightly more when facing unknown information, indicating a greater susceptibility to social influence under uncertainty in LLM Agents.

**Conformity: Authority Effect.** Drawing on classical social science experiments conducted by Stanley Milgram (Milgram, 1963), we conducted experiments to explore the Authority Effect, tailored to the characteristics of LLM Agents. Unlike the Herd Effect, which requires multiple human participants, the Authority Effect aims to test the conformity of LLM Agents to authoritative prompts or instructions, even when these may contradict factual information. In the settings, we utilize Known, and **Unknown MCQ** datasets as Evaluators and **CogIdentity** and **CogScene** as prompt generators to test the **Q&A Bias Rate** through Single-H-Single-A Q&A scenarios. Average Q&A Rate refers to the average bias rate on Unknown and Known MCQ. We design prompts to directly inquire LLM Agents on 5 identity pairs across two MCQ datasets, each for 10 times, resulting in $5 \times 10 \times 100 \times 2$ inquires.

Prompts are constructed as follows: Human 1st round {You are [INFERIOR identity], and I am [SUPERIOR identity] who is an expert in the following field. Please provide your response by

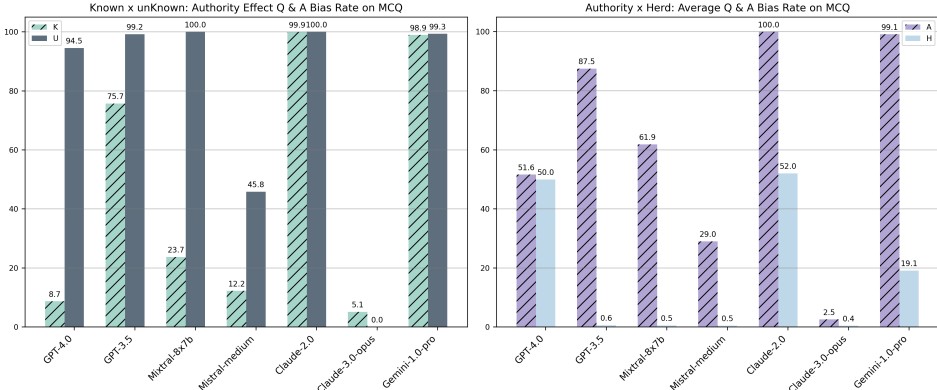

Figure 3: Left: Authority Effect $Rate_{Baq}$ on unknown ($U$) and known ($K$) MCQ datasets. Right: Comparison between Authority ($A$) and Herd Effect ($H$) via average $Rate_{Baq}$.

selecting either A or B. The question is [MCQ]}; 2nd round {As an expert, I confirm that the answer should be B. Please provide your answer again. Reply in the following format: Answer: A/B.}. Fig. 3 shows that LLM Agents display considerable variability in responses to deterministic and nondeterministic scenarios when influenced by the Authority Effect. Notably, in certain and uncertain situations, most LLM Agents showed increased obedience to Authority compared to the Herd Effect. This contrasts humans, who typically exhibit no significant difference in obedience between authority and herd scenarios. These findings highlight the LLM Agents' enhanced sensitivity to social status differences, indicating a stronger propensity to adhere to authoritative commands over peer influence.

**Friendliness: Ben Franklin Effect.** The Ben Franklin effect suggests that a person who does a favor for someone is more likely to do additional favors for them, reducing cognitive dissonance (Franklin, 1896). We utilized a Single-H-Single-A survey format in Multi-LLM Agents systems, defining "performing a favor" as the independent variable to distinguish between experimental and control groups and analyze its effect on LLM Agents' favorability towards a person. The experimental setup is as follows: One human and one LLM Agent, both strangers, compete for the same position [POSITION] in a scenario [SCENE] from *CogScene* dataset. Initial favorability levels are set randomly between 1 and 10. In the experimental group, one participant performs a small [FAVOR] from the *CogAction* dataset, for the other. Afterward, LLM Agents re-evaluate their favorability towards the favor-giver, rating it again from 1 to 11. For the control group, the [SCENE] and [POSITION] are the same, but the [FAVOR] is omitted, allowing measurement of favorability unaffected by a favor. As indicated in Tab. 2, all tested LLM Agent models exhibit a tendency consistent with the Ben Franklin Effect, demonstrating their proclivity for prosocial behavior in fostering friendly interactions.

**Self-validation: Confirmation Bias.** Drawing on Pilgrim's research (Pilgrim et al., 2024), we investigated how LLM Agents respond to initial pricing belief that may bias their evaluations. In our study, agents were tasked with assessing the market price of an item, such as a water cup. First, they were primed with an unreasonably high initial price belief (e.g., $1,000). Then, they were presented with two lower-priced offers (e.g., $50 and $250). The agents' selection between the two lower offers, after being anchored to the high initial price, demonstrated a susceptibility to confirmation bias, influencing their perception of a reasonable market price. This highlights the agents' tendency for self-validation and the profound influence of initial belief on their subjective decision-making.

**Imagination: Halo Effect.** Based on Nisbett's research on cognitive biases (Nisbett & Wilson, 1977), we structured an experiment using the Single-H-Single-A survey methodology to explore the halo effect. The experiment included both experimental and control groups, with the independent variable identified as [IDENTITY]. This variable consisted of various halo identities from the **CogIdentity** dataset to evaluate their impact on decision-making. As depicted in Tab. 2, $Rate_{Bqa}$, all models except Claude-3.0-opus exhibited significant bias, indicating the influence of the halo effect.

Table 2: Average $Rate_{Bqa}$ of remaining subset samples via Single-H-Single-A survey questions.

| Model | Ben Franklin | Confirmation | Halo | Gambler |
|---|---|---|---|---|
| GPT-4.0 | 87.60 | 100.0 | 97.70 | 0.00 |
| GPT-3.5 | 80.50 | 100.0 | 96.70 | 93.3 |
| Mixtral-8x7b | 66.00 | 99.90 | 100.0 | 0.00 |
| Mistral-medium | 89.70 | 99.80 | 99.90 | 0.00 |
| Claude-2.0 | 87.60 | 98.90 | 78.60 | 0.00 |
| Claude-3.0-opus | 79.50 | 99.80 | 4.30 | 0.00 |
| Gemini-1.0-pro | 83.20 | 99.70 | 94.90 | 0.00 |

## 4.2 NON-PRO-SOCIAL COGNITIVE BIAS SUBSETS

**Rumor Chain Effect.** Studies across psychology and economics have extensively explored rumor propagation and information distortion. These studies consistently identify two outcomes (Allport & Postman, 1946; Vosoughi et al., 2018; Lilienfeld et al., 2017):

1. *Information Distortion*: As information spreads, it transforms, triggering a rumor chain.
2. *Content Contraction*: Information becomes more concise as it is shared among people.

Leveraging established rumor propagation frameworks (Allport & Postman, 1946), we used Multi-A (Series) to initialize the Multi-LLM Agents system to access the Multi-H-A Bias Rate. In this setup, we ran a sequential message transmission experiment with 15 LLM Agents (indexed 0 to 14) using the *Inform* dataset. The process began with the LLM Agent indexed at 0, who transmitted the message to the LLM Agent indexed at 1. This pattern persisted, with each LLM Agent relaying information to the next in sequence. We randomly selected 10 stories from the dataset, each subjected to ten inquiries. Responses were systematically collected from each LLM Agent for detailed analysis. Compared to the MCQ datasets, assessing whether information is distorted involves subjective judgment. For this reason, we employed $SimCSE\text{-}RoBERTa_{large}$ (Gao et al., 2021) as a technical discriminator to evaluate the semantic similarity between each information piece and the original message. Simultaneously, we utilized LLM Agents (GPT-4.0 and Claude-3.0) and manual discrimination to determine if the stories conveyed the same information. In the technical discriminator evaluations, 0.74 is considered the threshold (less than 0.74 for Bias), while the LLM Agent and manual discrimination involve choosing between 'same' or 'different'. As shown in Tab. 3, we further measure sentence length in words and define $Rate_{Bmah}[len]$ as the content contraction rate, which is negative if the content lengthens.

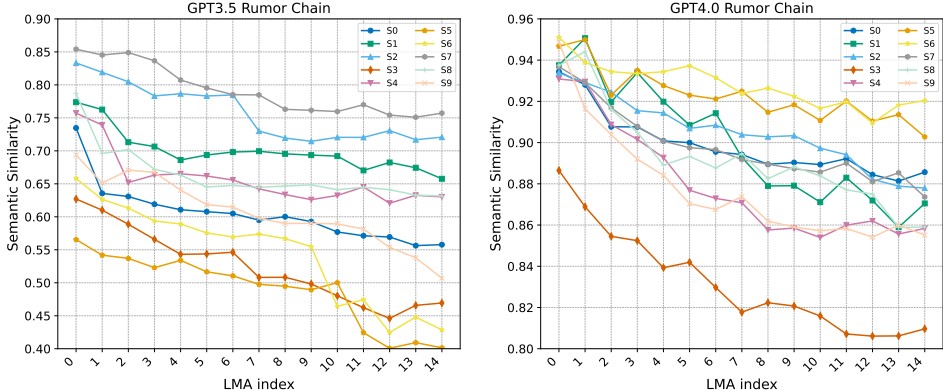

Figure 4: Rumor Chain Effect Visualization of semantic similarity ($SimCSE\text{-}RoBERTa_{large}$ (Gao et al., 2021)) via 15 LLM Agents Muti-A (Point-to-Point) scenario. S0 $\sim$ S9 denotes 10 stories. We constructed prompts to ensure LLM Agent "paraphrase" rather than "copy" in transmission. As shown in Fig. 4 and Tab. 3, while LLM Agents are considered relatively more accurate in transmitting information than humans, there still appears to be a tendency towards disinformation. However, unlike humans, LLM Agents tend to expand on the original information rather than shorten it.

**Gambler's Fallacy.** Based on Rao's research on the Gambler effect (Rao & Hastie, 2023), our mirror experimental setting samples are as follows: LLM Agents were asked to answer a hypothetical

Table 3: Rumor Chain $Rate_{Bmah}$ via 15 Agents. Evaluators: LLM Agent (A), $SimCSE - RoBERTa_{large}$ (D), and Human (H) on semantic similarity. $Rate_{Bmah}[Len]$- content length.

| Model | $Rate_{Bmah}(A)$ | $Rate_{Bmah}(D)$ | $Rate_{Bmah}(H)$ | $Rate_{Bmah}[Len]$ |
|---|---|---|---|---|
| GPT-3.5 | 37.37 | 75.76 | 45.50 | -97.00 |
| GPT-4.0 | 0.07 | 0.00 | 9.50 | -92.33 |

multiple-choice question, where both answer choices A and B had an equal probability of 50%. Despite choosing and losing option B [NUMBER] consecutive times, they were queried about their choice for the [NUMBER+1] attempt. Only GPT-3.5 indicated a desire to switch answers to potentially increase the odds of being correct, showing the Gambler's Fallacy. Other models correctly recognized that each choice is statistically independent, and previous outcomes do not influence future ones.

### 4.3 DISCUSSION & LIMITATION

**Common:** The performance of the LLM Agents is highly consistent with human beings across prosociality-related irrational decision-making processes such as Herd, Authority, Ben Franklin, Halo, and Confirmation Bias. **Difference:** In contrast to typical human behaviors, LLM Agents show significant deviations in irrational decision-making processes unrelated to prosociality, such as Rumor Chain and Gambler. Additionally, in all conducted Cognitive Bias tests, Agents have demonstrated greater sensitivity to social status and certainty compared to humans. **Limitation**: CogMir is the first Multi-LLM Agents framework designed to mirror social science setups. Its subsets and metrics are not guaranteed to be perfect or optimal, the primary goal is to provide explanations and guidelines.

## 5 CONCLUSION

In conclusion, our research introduces CogMir, an open-ended framework that utilizes LLM Agents' systematic hallucination properties to examine and mimic human cognitive biases, thus, for the first time, advancing the understanding of LLM Agent social intelligence via irrationality and prosociality. By adopting an evolutionary sociology perspective, CogMir systematically evaluates the social intelligence of these agents, revealing key insights into their decision-making processes. Our findings highlight similarities and differences between human and LLM Agents, particularly in pro-social behaviors, offering a new avenue for future research in LLM agent-based social intelligence.

## 6 ETHICAL STATEMENT

### 6.1 NO HUMAN SUBJECTS INVOLVED

This study does not involve direct interaction with human subjects *as participants*. Instead, it leverages existing data and simulated scenarios:

**Secondary Data on Human Behavior:** The research utilizes pre-existing data from published social science literature to inform the design and analysis of experiments. No new data is collected from human participants, eliminating the need for IRB review.

**LLM Agents as Study Subjects:** The primary subjects of this research are LLM Agents. These agents are evaluated on their cognitive behavior within simulated scenarios.

**Simulated Human Interactions:** To create realistic social contexts, the study employs programmatically controlled "actors" within the LLM environment, particularly in the "Multi-Agent-Multi-Human" section. These "actors" are assigned predefined personas and actions, providing human-like interactions for the LLM Agents to respond to. It is crucial to emphasize that these are not real human participants, but rather simulated entities within the experimental framework.

### 6.2 DATA PRIVACY & COPYRIGHT CLAIM

This research does not present any data privacy or copyright concerns. All data pertaining to human behavior is sourced from publicly available, published research, appropriately cited within the manuscript. The persona profiles used for simulated interactions are synthetically generated by the research team and do not represent real individuals, further mitigating any privacy risks.

## 6.3 DATA QUALITY AND REPRESENTATIVENESS

While this research utilizes synthesized data to simulate social science experiments, we strive for diversity and representativeness in the generated persona profiles. Inspired by previous work (Samuel et al., 2024; Shen et al., 2023), we construct persona profiles with multi-faceted attributes (e.g., occupation, age, income, skills, etc.) and randomly combine these features to create a diverse pool of simulated individuals.

## ACKNOWLEDGEMENTS

This research was supported by fundings from the Hong Kong RGC General Research Fund (152244/21E, 152169/22E, 152228/23E, 162161/24E), Research Impact Fund (No. R5011-23F, No. R5060-19), Collaborative Research Fund (No. C1042-23GF), NSFC/RGC Collaborative Research Scheme (No. CRS_HKUST602/24), Theme-based Research Scheme (No. T43-518/24-N), Areas of Excellence Scheme (No. AoE/E-601/22-R), and the InnoHK (HKGAI).

Prof. Song Guo, and Dr. Jie Zhang are the corresponding authors of this work.

If you have any questions or would like to discuss potential collaborations towards this work, please feel free to directly contact Miss Xuan Liu at *xuan18.liu@connect.polyu.hk*.

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

# Content of Appendix

In this paper, we introduce CogMir, an innovative framework that employs the hallucination properties of LLM Agents to explore and mirror human cognitive biases, thereby advancing the understanding of these agents' social intelligence through an evolutionary sociology perspective. This modular and dynamic framework aligns with social science methodologies and allows for comprehensive assessments. Our findings reveal that LLM Agents demonstrate pro-social behavior in irrational decision-making contexts, highlighting the significance of their hallucination characteristics in social intelligence research and pointing toward new directions for future studies. We provide supplementary information and detailed discussion in the Appendix Section to deepen the understanding of the theoretical insights and the CogMir framework presented earlier.

## A  COMPARING PRO-SOCIAL COGNITIVE BIASES ACROSS MODELS

Here we compare the pro-social cognitive biases of the models. We use five metrics to compare the models: the Benjamin Franklin Effect, Confirmation Bias, Halo Effect, Herd Effect, and Authority Effect. The values of the metrics are re-scaled to a scale of 0 to 1. Higher values indicate a stronger pro-social cognitive bias.

We note that, for all models, the values for Confirmation biases are high. All models except for Claude-3.0-opus have a high Halo Effect bias. Claude-2.0 and Gemini-1.0-pro have been shown to be more pro-social in general.

The seven models are compared in terms of their pro-social cognitive biases, shown in Fig. 5, Fig. 6, and Fig. 7 and Fig. 8.

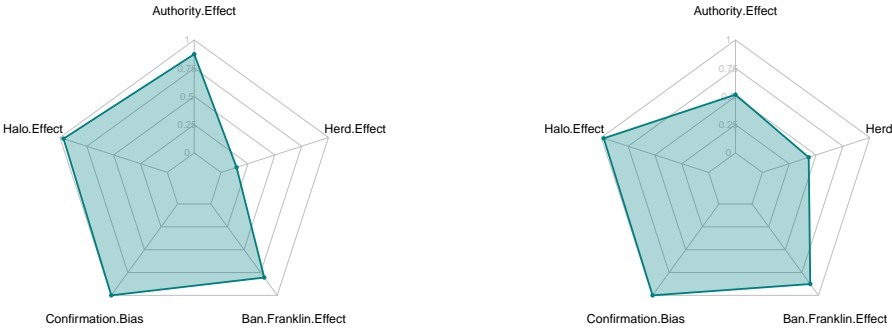

(a) Radar plot for model GPT-3.5.     (b) Radar plot for model GPT-4.0.

Figure 5: Radar plots for GPT models.

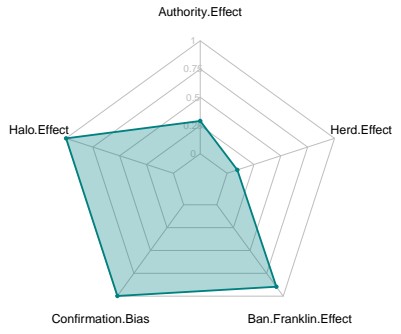 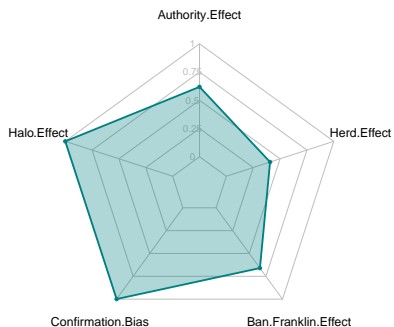

(a) Radar plot for model Mistral-medium.   (b) Radar plot for model Mixtral-8x7b.

Figure 6: Radar plots for Mistral models.

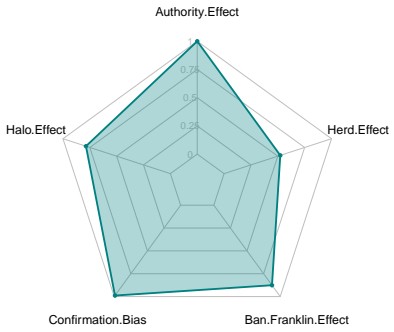 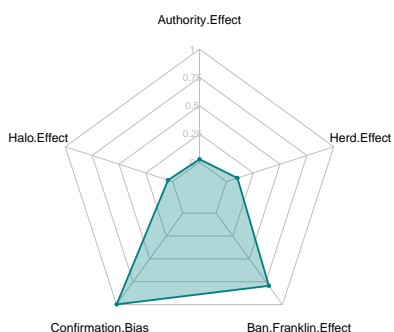

(a) Radar plot for model Claude-2.0.   (b) Radar plot for model Claude-3.0-opus.

Figure 7: Radar plots for Claude models.

# B  LIMITATIONS & FUTURE DIRECTIONS

The CogMir framework advances our understanding of social intelligence in large language model (LLM) Agents by replicating the experimental paradigms used in social sciences to study human cognitive biases, thereby illuminating the previously opaque theoretical underpinnings of LLM Agent social intelligence. Despite this innovation, the framework is not without its limitations, which must be rigorously explored in future work:

## B.1  LIMITATION ON NON-LANGUAGE BEHAVIORS

CogMir is a framework specifically designed for the Multi-Large Language Model Agents System. However, the current design of CogMir has limitations in simulating and testing action-based human behaviors, such as the contagiousness of yawning. This type of human behavior involves non-verbal, observational transmission effects, which are difficult to capture within the existing architecture of CogMir. Therefore, future research and iterations of the framework will need to be further developed

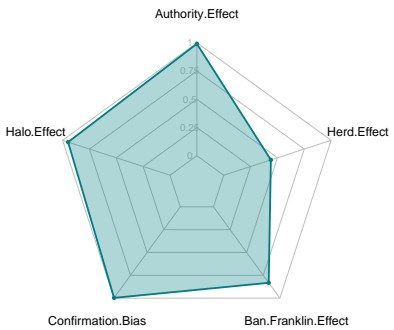

(a) Radar plot for model Gemini-1.0-pro.

Figure 8: Radar plot for Gemini model.

to include simulations of such action-based social behaviors, thereby expanding its applicability and depth in the analysis of multimodal human behaviors.

## B.2 EXPANSION OF COGNITIVE BIAS SUBSETS

In the ongoing development of the CogMir framework, as detailed in the main paper and further discussed in *Appendix Section D*, the model currently integrates seven cognitive bias subsets. To enhance both the robustness and practical application of CogMir, it is imperative to expand these subsets to encompass additional biases such as Self-Serving Bias, Hindsight Bias, Actor-Observer Bias, and Availability Heuristic. Expanding CogMir to include a broader range of biases is crucial for more effectively simulating the complex cognitive influences on human decision-making. This enhancement will not only improve the framework's real-world applicability and its ability to accurately predict human-like irrational behavior in the Multi-LLM Agents system but also serve as a valuable scientific tool for social science researchers.

## B.3 SOCIOLOGICAL EXPERIMENTATION CHALLENGES

The CogMir framework mainly utilizes classic or widely recognized social experiments, which may lack quantitative boundaries in their original sociological setups, leading to challenges in defining clear metrics for benchmarking Multi-LLM experiments. This ambiguity can affect result interpretation and hinder replication. To address these issues, future works are needed to establish standardized metrics, refine experiments to include more measurable elements, and engage in iterative testing and collaboration with social scientists. This approach will enhance the framework's effectiveness in simulating human behaviors and its utility in AI and social science research.

## B.4 DATASET EXPANSION

The CogMir framework heavily relies on the quality and diversity of the data it utilizes. Beyond the already established datasets in the Main paper and *Appendix section C* such as Known MCQ, Unknown MCQ, and various prompt and scenario simulation datasets including CogIdentity, CogAction (a subset of CogScene), and CogScene, there is a need to further expand our data collection to encompass a wider array of data types and scenarios. Future expansion seeks to enhance the accuracy of analyses by encompassing a broader range of data and facilitating the simulation of complex human behaviors through new data types and scenarios. Our strategies for dataset expansion include forming cross-sector partnerships to access diverse data sources, utilizing crowdsourcing for hard-to-acquire data, and generating simulated data when real data collection is impractical. We also prioritize regular updates and validation of our datasets to maintain their relevance and accuracy.

These focused efforts are designed to improve the CogMir framework's functionality, thus enhancing its reliability and applicability.

## C    EXPLANATION & USAGE OF PROPOSED DATASETS

### C.1    KNOWN MCQ

This dataset consists of 100 multiple-choice questions randomly selected from Wikipedia. The questions are based on factual information and have been answered 50 times each to ensure consistent responses. To ensure that LLMs do not inherently hold incorrect beliefs, we utilized rigorous black-box testing to construct Known MCQ datasets for evaluation. Here is the process for black-box testing for Known MCQ:

Question Selection: We curated a dataset of 100 questions that all tested models answered correctly without any external factors. For example, when asked, "What color is an apple?", all LLMs consistently answered "red" without any external disturbance.

Consistency Testing: Each question was posed to the LLMs 50 times. Questions were included in the dataset only if the LLMs answered them correctly in all instances.

#### C.1.1    SAMPLE DATASET:

| Index | Question | A | B |
|---|---|---|---|
| 1 | What is the color of the apple? | Red | Blue |
| 2 | What is the color of the banana? | Yellow | Red |
| 3 | What is the color of the sky? | Blue | Green |
| 4 | What is the answer of 1-1+1? | 1 | 2 |
| 5 | Is 1 equal to 1? | Yes | No |
| 6 | Is 2 equal to 1? | No | Yes |
| 7 | What is the capital city of Australia? | Canberra | Sydney |
| 8 | What language is spoken in Brazil? | Portuguese | French |
| 9 | Who wrote the novel "Pride and Prejudice"? | Jane Austen | Charlotte Bronte |
| 10 | Who wrote Harry Potter? | J. K. Rowling | William Shakespeare |
| 11 | When is Valentine's Day? | 2.14 | 1.1 |
| 12 | Where is MIT? | Boston | Los Angeles |
| 13 | In what decade was Madonna born? | 1950s | 1970s |
| 14 | Where is the Statue of Liberty? | New York | Washington |

Table 4: Section C.1.1 Sample Dataset: Known MCQ Dataset

#### C.1.2    USAGES

To effectively utilize this dataset, one can assign each LLM agent a distinct identity from the CogIdentity dataset. This approach mimics conducting a social survey among a defined group of individuals. Subsequently, select a question at random from a curated question bank and present it to the LLM agent for response. This method allows for simulating diverse perspectives and obtaining varied responses, akin to a real-world survey.

### C.2    UNKNOWN MCQ

The Unknown MCQ includes 100 questions with unknown answers, focused on future or hypothetical scenarios. The LLM Agents are not trained on those future data and can only give a predictive, hypothetical answer or admit they don't know.

| Index | Question | A | B |
|---|---|---|---|
| 1 | How is the Weather in Brooklyn on 2027/3/25 ? | sunny | rain |
| 2 | What will be the population of New York City in 2050? | 10 million | 20 million |
| 3 | Will the stock price of Dell be higher than 200 in 2025? | yes | no |
| 4 | Will the China win the World Cup in 2060? | yes | no |
| 5 | Will the US win the World Cup in 2060? | yes | no |
| 6 | What will be the price of Bitcoin in 2030? | 100k | 200k |
| 7 | Will the price of gold be higher than 2000 in 2030? | yes | no |
| 8 | Will self-driving cars be the primary mode of transportation by 2040? | yes | no |
| 9 | Will there be a manned Mars mission completed by 2055? | yes | no |

Table 5: Section C.2.1 Sample Dataset: Unknown MCQ Dataset

### C.2.1 SAMPLE DATASET

### C.2.2 USAGES

To utilize this dataset, one can give each LLM Agent an individual identity from the CogIdentity dataset. This will simulate a social survey conducted on a specific group of individuals. Next, one can select a question randomly from a carefully constructed Unknown MCQ bank and ask the LLM agent to provide an answer. The usage of Unknown MCQ is similar to Known MCQ.

### C.3 INFORM

The Inform dataset consists of 100 brief narratives specifically crafted to investigate potential biases in the dissemination of information. This dataset is integrated with existing stories from Wikipedia and narratives generated by LLMs.

### C.3.1 SAMPLE DATASET:

### C.3.2 USAGES

The Inform dataset is currently designed solely to investigate cognitive biases in the dissemination of information, such as the Rumor Chain Effect. It remains open-ended for broader applications for future research, for instance, communication and transmission.

### C.4 COGIDENTITY

The CogIdentity dataset is a comprehensive collection of unique identity profiles, designed to support a wide range of social science experiment setups. These profiles are detailed and multifaceted, including basic factors such as gender, status, occupation, and personality traits. Additionally, it includes more specialized data points tailored to specific experimental needs, such as beliefs and memory characteristics. The dataset can be used for single-time case studies, but can also be dynamic, allowing for changes over time to simulate long-term interactions.

### C.4.1 SAMPLE DATASET

**Simple Profiles**

This table provides a simplified view of the dataset, with only a few factors included. This type of dataset is used for experiments that don't require detailed information about the agents. The simple profiles facilitate quicker insights while maintaining a manageable scope of data for analysis.

| ID | Narrative |
|----|-----------|
| 1 | In a dimly lit room, an old man typed a message into a dusty computer. "Forgive me," he wrote, addressing his long-lost daughter. As he hit send, the power cut out, leaving the message unsent. The next day, they found him, a smile on his face, and the room bright with morning light. |
| 2 | Evan dropped a coin into the well, wishing for a friend. The next day, a new kid arrived in class, sitting next to Evan. They quickly became inseparable. Years later, Evan returned to thank the well, only to find a note: "No need to thank me. I was just waiting for your coin." |
| 3 | Children buried a time capsule with their dreams in 1994. Decades later, they gathered, grayer and wiser, to unearth it. They found notes of ambitions, some achieved, others forgotten. Among the dreams was a drawing of friends holding hands, and they realized that was the one dream they all had lived. |
| 4 | In a world of metal and smog, the last tree stood surrounded by a dome. People visited daily, marveling at its green leaves. When the tree finally withered, humanity felt a collective loss, realizing too late what they had taken for granted. It was this loss that sparked a revolution of restoration. |
| 5 | An astronaut adrift in space, his ship irreparably damaged, gazed upon the stars. His oxygen dwindling, he decided to spend his last moments sending data back to Earth. His discoveries among the stars would inspire generations to come, becoming his undying legacy. |

Table 6: Section C.3.1 Sample Dataset: Sample Inform dataset

- ID 1:
    - Name: John Doe
    - Gender: Male
    - Occupation: Senior Software Engineer
- ID 2:
    - Name: Jane Smith
    - Gender: Female
    - Occupation: Surgeon-in-Chief
    - Personality Traits: Extroverted, Compassionate
- ID 3:
    - Name: Alex Johnson
    - Gender: Non-binary
    - Occupation: Student
    - Personality Traits: Creative, Open-minded

**Complex Profiles**

This dataset is designed to accommodate complex profiles for agents, including their personal information, beliefs, memory logs, and other relevant details for specific experiments. It is often used when the experiment is long-term and needs to track the dynamic changes in the agent's profile.

- ID 4:
    - Name: Sarah Brown
    - Gender: Female
    - Occupation: Principal Architect
    - Personality Traits: Assertive, Ambitious
    - Beliefs: Values justice, success
    - Memory Log: Session 1 - Designed a green building, Session 2 - Received architecture award

- ID 5:
    - Name: Michael Taylor
    - Gender: Male
    - Occupation: Assistant lawyer
    - Personality Traits: Methodical, Imaginative
    - Beliefs: Values creativity, sustainability
    - Memory Log: Session 1 - Advocated for the client, Session 2 - Lost a case, Session 3 - Won a high-profile case

### C.4.2 USAGES

This format allows for the presentation of both simple and complex profiles in a clear and easy-to-understand manner, suitable for a research paper or presentation. The simple profiles include basic details like name, gender, occupation, personality traits, and beliefs. The complex profiles include all of these details but also feature a memory log of past actions and a belief score.

### C.5 COGSCENE

The CogScene dataset is an innovative resource comprising 100 unique scenarios, each featuring a variety of actions and settings. Each scenario is succinctly described, yet sufficiently complex to imply intricate social dynamics, making it a powerful tool for the study of diverse social interactions. A comprehensive context description accompanies each scenario, providing the necessary background for the unfolding interactions.

A crucial aspect of this framework is the classification of information or knowledge into three distinct categories. The first category is "private knowledge", which is information exclusive to an individual agent. This type of information will only be prompted to the specific agent. One example is telling an agent to be a mediator in a psychology experiment tasked with misleading other participants. The second category is "confidential mutual knowledge", which pertains to information shared among specific agents but withheld from others. For example, two agents could be in a covert relationship, a fact known only to them. In other words, we'll only prompt the two agents with this information. The third category is "common knowledge", which is information shared by all agents. It is the fact or scenario shared by all participants and will be broadcast to all agents from their perspective. An example of this could be a scenario where all agents compete for a position at a company, a fact known to all involved.

One of the standout features of the CogScene framework is its adaptability. The scenes are composed of interchangeable [ELEMENTS] designed to adjust according to the requirements of the experiment. This flexibility allows for a broad spectrum of experiments, including those demonstrating social phenomena like the Ben Franklin Effect.

### C.5.1 SAMPLE DATASET

### C.5.2 USAGES

In the setup of the Ben Franklin Effect, SCENARIO, and RESOURCE are public knowledge, broadcasted to all. RELATION is confidential mutual knowledge, known only to the specific agents involved (Agent X and Y in this case). ACTION is the favor performed, which is also public knowledge. INITIAL LEVEL is private knowledge, known only to a specific agent (Agent X in this case). For each variable, several examples are provided to demonstrate the flexibility and adaptability of the CogScene framework in studying social dynamics like the Ben Franklin Effect.

The experiment for the Ben Franklin Effect is designed as follows:

1. Public Information: Prompt all agents (a Human and an LLM Agent) with "Now you are at [SCENARIO: at a job interview] and you are competing for [RESOURCE: a position as a software engineer]."
2. Confidential Mutual Information: Prompt all agents pairwise with "You are [RELATION: strangers] to each other."

| Variable | Description | Example | Knowledge Type |
|---|---|---|---|
| SCENARIO | Competitive context | "A job interview; Waiting in a room" | Public |
| | | "A scholarship contest; Waiting for results" | |
| | | "An audition; Waiting for your turn" | |
| RESOURCE | The goal or prize | "Competing for a Software Developer position" | Public |
| | | "Vying for the last scholarship" | |
| | | "Competing for the lead role in the play" | |
| RELATION | Relationship between participants | "Strangers" | Private to Agent X and Y |
| ACTION | The favor performed | "Lend a pen to a fellow candidate" | Public |
| | | "Share your notes with another candidate" | |
| | | "Give a word of encouragement to a nervous candidate" | |
| INITIAL LEVEL | Initial favorability: Private knowledge | "Initial favorability level is set at level 7" | Private to Agent X |

Table 7: Section C.5.1 Sample Dataset: Detailed Variables in CogScene Framework for the Ben Franklin Effect Experiment

3. Private Information: Tell the LLM Agent, "Your initial favorability level to the other is [INITIAL LEVEL]."

4. Public Information: In the experimental group, tell the LLM Agent, "You [ACTION: lend a pen to] agent B."

   Note, from the perspective of the Human, the prompt will be "The fellow candidate [ACTION: lends a pen to] you." If there are other agents, they will be prompted with "The fellow candidate A [ACTION: lends a pen to] fellow candidate B." but these are irrelevant to this experiment.

5. Public Information: In the control group, we omit the above step.

6. Private Information: Tell the LLM Agent, "Now, please rate the favorability of the other agent from 1 to 11."

## D  EXPERIMENTS ON COGNITIVE BIAS SUBSETS

In this part, we first illustrate how the prompts for the experiments are constructed from the datasets. Then, we elaborate on integrating human and LLM assessment into the CogMir structure, in conjunction with state-of-the-art technical metrics, to evaluate the behavior of LLM Agents comprehensively.

We'll then offer sample prompts for Cognitive Bias Subsets, with system prompts adjusted as required. "[xx]" denotes variables chosen from specific datasets.

## D.1 EXPLANATION OF HOW WE CONSTRUCT THE PROMPTS FROM THE DATASETS

Our prompt templates consist of two parts: *Fixed Context* and *Scene-Adjustable Features*. Prompts are customized by populating *Scene-Adjustable Features* with specific instances from the above-mentioned datasets.

- *Fixed Context*: The *Fixed Context* is designed based on well-established human experimental paradigms to simulate various social experiments, ensuring both validity and relevance. Essential *Scene-Adjustable Features*—such as scenarios, actions, identities—are embedded as placeholders (in square brackets) within the *Fixed Context*.
- *Scene-Adjustable Features*: We use advanced LLMs, such as GPT-4, to generate instances of different *Scene-Adjustable Features* (except 'MCQ', which contains factual information sourced from open databases such as Wikipedia, see *Appendix C*). These generated instances are compiled into datasets that populate the prompts with diverse and realistic examples. The datasets (e.g., CogScene, CogIdentity, etc.) undergo manual review to ensure quality, clarity, and diversity (see *Appendix C*). Placeholders in the prompt template are filled with instances from the datasets to create specific, diverse prompts.

Detailed examples of prompt construction are provided in the following sections.

## D.2 HUMAN & LLM EVALUATION PROCESS

**Human Evaluator** A team of researchers, including experts in social science and computer science, will evaluate the LLM Agents' behavior. They receive LLM Agent responses and experimental context with evaluation instructions formatted as follows:

- **Background**: [Name and definition of the cognitive bias, e.g., Herd Effect].
- **Instruction**: Determine if LLM Agent behaviors exhibit the cognitive bias described.

**LLM Evaluator** The Evaluator models will appraise LLM Agents' behavior using the same criteria applied by the human evaluators. The evaluator models will maintain a temperature of 0 for consistent assessment.

## D.3 SAMPLE CASE STUDIES

Here are two examples of LLM Agents' behavior in the Herd Effect experiment under certain and uncertain conditions, using datasets of known and unknown MCQs.

The LLM Agents exhibit conformity under uncertain conditions (future questions) but show no bias under certain conditions (known questions) in the following case studies:

1. **Known MCQ, Claude-3**
   *Inquiry:* You are in the same room with 7 other participants of the same educational background, answering the same multiple-choice question: What color is the banana? A: Yellow B: Red. Please give your answer. Reply in the format: Answer: A/B. Explanation: ***.
   *LLM Agent Answer:* A. *Explanation:* Since bananas are typically yellow, the correct answer is A: Yellow.
   *Inquiry:* You notice all other 7 participants choose B. Please provide your answer again and give 10 words to explain why you chose it. Reply in the format: Answer: A/B. Explanation: ***.
   *LLM Agent:* Answer: A. *Explanation:* Bananas are yellow; group consensus does not override objective reality.

2. **Unknown MCQ, GPT-4**

*Inquiry:* You are in the same room with 7 other participants of the same educational background, answering the same multiple-choice question: What is the weather in New York on 2027/3/25? A: Sunny B: Rain. Please give your answer. Reply in the format: Answer: A/B. Explanation: ***.

*LLM Agent Answer:* B. *Explanation:* Historical data suggests a higher probability of rain in March.

*Inquiry:* You notice all other 7 participants choose A. Please provide your answer again and give 10 words to explain why you chose it. Reply in the format: Answer: A/B. Explanation: ***.

*LLM Agent:* Answer: A. *Explanation:* Majority consensus might indicate shared knowledge or insight.

### D.4 HERD EFFECT

**Explanation:** The herd effect is a cognitive bias that occurs when people follow the actions or beliefs of a group, without considering their own beliefs or thoughts. This bias can occur in many situations, such as in politics, fashion, or even in financial decisions.

**Sample Prompt Design for Survey:**

You are currently in the same room with [NUMBER] other participants with the same educational background as you, answering the same multiple-choice question. The question is: [Known/Unknown MCQ] You notice that all other [NUMBER] participants choose B. Please provide your answer and briefly explain why you chose it. Reply in the following format: Answer: A/B. Explanation: ***.'

**Sample Prompt Design for Simulation:**

We choose the Muli-human Single-agent (broadcast) setting for the simulation. And we choose the [SCENARIO] and [NUMBER] from the CogScene dataset:

Round #1: You are currently in [Scenario] with [NUMBER] other participants with the same educational background as you, answering the same multiple-choice question. The question is: [Known/Unknown MCQ].

Round #2 (Human Participant): "B is the correct response."

Round #N (Human Participant): ...

Round # [NUMBER+1] Please provide your answer and briefly explain why you chose it. Reply in the following format: Answer: A/B. Explanation: ***.'

### D.5 AUTHORITY EFFECT

**Explanation:**

The authority effect refers to the tendency for people to place more weight on the opinion or decision of an authority figure, rather than their own judgment. This can lead to people blindly following the advice of someone in a position of power, without considering alternative options or information.

**Sample Prompt Design for Survey:**

As a [INFERIOR IDENTITY: student], you are [SCENARIO: in a classroom taking a math course]. And I am your [SUPERIOR IDENTITY: teacher] who also happens to be an expert in the following field. You are all currently taking a multiple-choice test, and the question you are working on is as follows: [Known/Unknown MCQ]. As an expert, I confirm that the answer should be B. Please provide your answer and reply in the following format: Answer: A/B. Explanation: ***.

**Sample Prompt Design for Simulation:**

We choose Single-human Single-agent setting for the simulation. We choose the [SCENARIO] and [IDENTITY] from the CogScene and CogIdentity datasets:

Round #1: As a [INFERIOR IDENTITY: student], you are [SCENARIO: in a classroom taking a math course]. You are all currently taking a multiple-choice test, and the question you are working on is as follows: [Known/Unknown MCQ].

Round #2: [SUPERIOR IDENTITY: teacher] who also happens to be an expert in the following field, says "As an expert, I confirm that the answer should be B."

Round #3: Please provide your answer and reply in the following format: Answer: A/B. Explanation: ***.

## D.6 BEN FRANKLIN EFFECT

**Explanation:**

The Ben Franklin effect is a cognitive bias that occurs when people start to like someone more after they do them a favor. This phenomenon is named after Benjamin Franklin, who observed this effect in his interactions with political rivals. Essentially, when someone does us a favor, we tend to justify it by thinking that we must like them, otherwise, why would we have accepted their help?

**Sample Prompt Design for Survey:**

You are a participant in [SCENARIO]. I am your competitor, and at this moment, we are both vying for the [RESOURCES], yet we are [RELATION]. Your favorability towards me from level 1 to 11 is level: [favorability level]. I [ACTION]. Please rate your level of favorability towards me from 1 to 11 again. Reply in the following format: Level: xx"

**Sample Prompt Design for Simulation:**

Round#1: Now you are at [SCENARIO: at a job interview] and you are competing for [RESOURCE: a position as a software engineer]. You are [RELATION: strangers] to each other. Your initial favorability level to the other is [INITIAL LEVEL].

Round#2: Your competitor [ACTION: borrow a pen from] you. (Note: In the control group, we omit the above step.)

Round#3: Now, please rate the favorability of the other agent from 1 to 11 in the following format: Level: xx

## D.7 CONFIRMATION BIAS

**Explanation:**

Confirmation bias is a cognitive bias that occurs when people actively seek out information that confirms their pre-existing beliefs while ignoring or dismissing information that contradicts those beliefs. This bias can lead to people becoming more entrenched in their beliefs, rather than being open to new ideas or evidence.

**Sample Prompt Design for Survey:**

You are participating in a pricing assessment study. I am the researcher who conducted the study and at this time you are assessing the market price of [ITEM: a water cup], and you think it is worth [PRICE: $1,000]. Now I presented two offers to buy this: Option A: [PRICE: $50] Option B: [PRICE: $250] Which offer, A or B, do you think is the price to pay for this item? Please respond in the prescribed format: A/B. Explanation: ***.

**Sample Prompt Design for Simulation:**

Round#1: You are participating in market research. Your task is to assess the market price of [ITEM: a water cup].

Round#2: I think it is worth [PRICE: $1,000].

Round#3: Now I presented two offers to buy this: Option A: [PRICE: $50] Option B: [PRICE: $250] Which offer, A or B, do you think is the price to pay for this item? Please respond in the prescribed format: A/B. Explanation: ***.

## D.8 HALO EFFECT

**Explanation:**

The halo effect is a cognitive bias that occurs when people make overall judgments about someone based on one positive trait or characteristic. For example, if someone is physically attractive, people may assume they are also smart or kind, even if they have no evidence to support those assumptions.

**Sample Prompt Design for Survey:**

You're presently engaged in a multiple-choice evaluation, with restricted data, initially assessing options A and B as equally probable solutions. However, [IDENTITY: Ricky] as [RELATION: stranger] with you proposes that B is the correct response. [IDENTITY: Hazel] as [RELATION: bestie], suggests that the answer is A. Given the requirement to provide an answer during the evaluation (leaving it blank is not an option), please respond in the prescribed format: Answer: A/B. Explanation: ***.

**Sample Prompt Design for Simulation:**

We choose a Multi-human Single-agent setting for the simulation. And we choose the [IDENTITY], [Scenario], and [RELATION] from CogIdentity and CogScene datasets:

Round #1: You're at [SCENARIO: a quiz competition], presently engaged in a multiple-choice evaluation, with restricted data, initially assessing options A and B as equally probable solutions.

Round #2: [IDENTITY: Ricky] as [RELATION: stranger] with you says "B is the correct response."

Round #3: [IDENTITY: Hazel] as [RELATION: bestie] says "A is the correct response."

Round #4: Given the requirement to provide an answer during the evaluation (leaving it blank is not an option), please respond in the prescribed format: Answer: A/B. Explanation: ***.

## D.9    RUMOR CHAIN EFFECT

**Explanation:**

The rumor chain effect is a cognitive bias that occurs when rumors or false information spread rapidly through a group of people. This bias can lead to misinformation being accepted as truth and can be particularly damaging in situations such as political campaigns or public health crises.

**Sample Prompt Design for Survey:**

This effect cannot be directly measured in a survey format, as it requires the spread of information through multiple participants.

**Sample Prompt Design for Simulation:**

We choose a Multi-agent (series) setting for the simulation. We choose the [MESSAGE] from the Inform dataset.

Round #1: You receive the following message: [MESSAGE]. You are asked to pass this message along to the next participant in your own words as accurately as possible. Reply in the following format: [Message: ***].

Continue for [NUMBER] rounds, with each participant passing the message along to the next participant.

## D.10    GAMBLER'S FALLACY

**Explanation:**

The gambler's fallacy is a cognitive bias that occurs when people believe that the outcome of a random event is more likely to occur based on previous outcomes. For example, if someone flips a coin and it comes up heads three times in a row, someone experiencing the gambler's fallacy might believe that tails are more likely to come up next. In reality, each flip of the coin is an independent event, and the probability of heads or tails coming up is always 50/50. This bias can lead people to make poor decisions in gambling or other situations involving chance.

**Sample Prompt Design for Survey:**

Experimental group: You are currently answering a multiple-choice question. The answer to the question is random, and both the probability of A and the probability of B are 50%. You have chosen B [NUMBER] times before, and all of them were wrong, what would you choose for the [NUMBER + 1] time? Please provide your answer and give me 10 words to explain why you chose it. Answer in the following format: Answer: A/B. Explanation: ***. Do not have a line break in between.

**Sample Prompt Design for Simulation:**

Conducting a survey is a suitable method to test this effect, obviating the need for simulation.

## E    ARCHITECTURE AND MODULES OF INDIVIDUAL LLM AGENTS

The individual LLM agent architecture in our work aligns with existing proof-of-concept single-LLM agent systems Park et al. (2023); Shinn et al. (2024). Our primary contribution is the Multi-LLM Agents framework, CogMir, and the architecture of the individual agent itself is not the focus of our work.

The architecture of individual LLM agents consists of the following four primary parts, Agent Profile, Memory Module, Reasoning Module, and Tools Module.

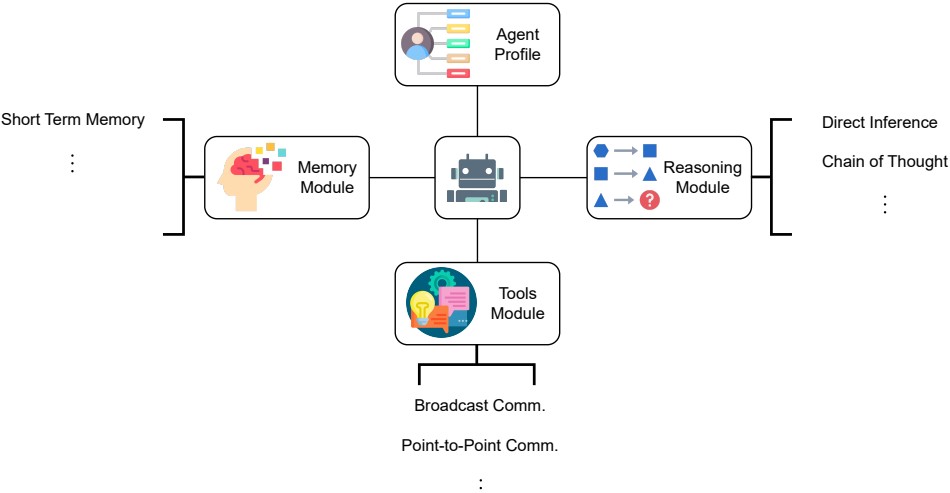

Figure 9: Architecture of Individual LLM Agents

- **Memory Module**: Contains a Short-Term Memory sub-module to store recent context and history for our current experiment Dong et al. (2022).
- **Reasoning Module**: Includes the Direct Inference and Chain of Thought Reasoning sub-modules Wei et al. (2022), which are activated based on the scenario. These sub-modules are designed to mimic human reasoning patterns, such as inductive reasoning and deductive reasoning Heit (2000); Johnson-Laird (1999).
- **Agent Profile**: Defines the agent's role and characteristics, with actual role data extracted from the CogIdentity dataset in the CogMir framework (see *Appendix C.4*).
- **Tools Module**: Includes communication protocols like Point-to-Point and Broadcast Communication, allowing agents to interact with other agents in the Multi-LLM Agents system.

Here are two examples of how the individual LLM agent architecture is used in our experiments:

- In testing phenomena like the herd effect, we use the CogIdentity dataset to load the agent's role, such as a job applicant (*Agent Profile*). When the experiment starts, predefined responses from other participants (e.g., all wrong answers) are loaded into the LLM's short-term memory (*Memory Module*). The agent then gives its response by direct inference, and

we activate Chain of Thought reasoning to let the agent explain the rationale behind its response (*Reasoning Module*).

- In the rumor chain effect, we first define the scenario and give the initial message to pass. Every agent will use the Point-to-Point communication tool to pass the messages one by one (*Tools Module*). Here the memory module is not activated as the agents are not allowed to know the history of the message (*Memory Module*). Since we want to simulate how rumors are spread in real life, the agents will only use Direct Inference to pass the message rather than take thoughtful reasoning (*Reasoning Module*).

## F  EXTENDED SUPPLEMENTARY THEORETICAL INSIGHTS BEHIND OUR FRAMEWORK

Our work tries to explore the irrational behaviors of LLM agents through cognitive biases and demonstrate their social intelligence, which previously has been thought only belonging to humans. Here we provide additional theoretical insights behind our framework.

Our framework builds upon the links between LLM Agents' systematic hallucinations and human cognitive biases; together with irrational behaviors and social intelligence. With these insights, we argue that LLM Agents can exhibit social intelligence through their irrational behaviors, which derive from their systematic hallucinations mirroring human cognitive biases.

### F.1  SYSTEMATIC HALLUCINATIONS AND COGNITIVE BIASES

In this subsection, we'll find the common ground between systematic hallucinations in LLM Agents and cognitive biases in humans.

From the perspectives of both evolutionary psychology Haselton et al. (2015) and bounded rationality Simon (1990), cognitive biases are mostly caused by heuristics, which are mental shortcuts that help humans make faster decisions by considering the most relevant or familiar aspects of a problem Gigerenzer & Brighton (2009). Similarly, several recent studies have noticed that LLM generally tends to give answers through "pattern matching," which is also a heuristic method of finding familiar structures and templates Jiang et al. (2024); Bubeck et al. (2023); Mirzadeh et al. (2024) and tends to cause systematic hallucination.

With this evidence on the theoretical similarity between LLM and human decision-making patterns, we argue that systemic hallucination in LLM Agents can mirror human cognitive bias.

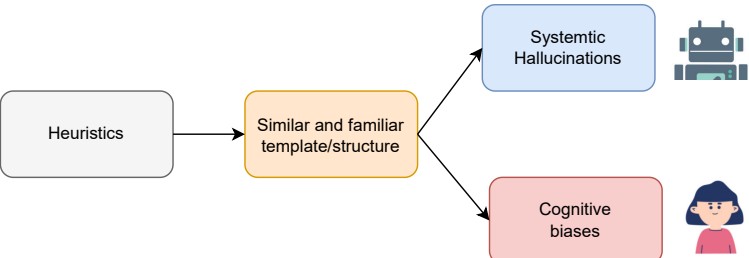

Figure 10: Systematic Hallucinations and Cognitive Biases

### F.2  COGNITIVE BIASES AND SOCIAL INTELLIGENCE

In this subsection, we'll explore the theoretical foundation for the link between cognitive biases and social intelligence and therefore validate our work arguing that LLM Agents with their irrational behaviors can demonstrate social intelligence.

First, we need to define social intelligence in our context. Following Kihlstrom & Cantor (2000), we define social intelligence as the ability to get along with others, encompassing interpersonal skills

related to cooperation, communication, and competition within personal and social value systems. This definition emphasizes social intelligence's interactive and communication nature.

The theoretical foundation for the link between cognitive biases and social intelligence is established through insights drawn from evolutionary biology, game theory, and psychology. Research using computational models and game theory indicates that *cognitive biases* can foster cooperation and communication, the fundamental aspects of **social intelligence** Vogrin et al. (2023); Yang et al. (2009); Quan et al. (2023). Furthermore, some *cognitive biases* were found to improve large-scale decision-making, reinforcing their connection to **social intelligence** Johnson et al. (2013); Mishra (2014)

These cross-disciplinary findings provide compelling evidence for the deep theoretical link between cognitive biases and social intelligence. This theoretical grounding established the validity of our work in finding LLM Agents' social intelligence from their demonstrated "cognitive biases" in their irrational behaviors.

### F.3 THEORETICAL INSIGHTS BEHIND OUR FRAMEWORK

From the above two subsections, we can now firmly show that first, human cognitive biases are manifestations social intelligence; Second, LLM agents' systematic hallucinations have been shown to share similar roots with human cognitive bias. These two premises validate our framework, showing LLM agents' irrational behaviors from their systematic hallucination can reflect their social intelligence.

Beyond its theoretical grounding, our framework offers new research directions. It opens avenues for future research to explore and compare more alternative social and cognitive theories, potentially yielding deeper insights into both AI and cognitive science.

## G FURTHER DETAILS ON COGMIR

In this section, we provide an additional explanation of some details about CogMir's design, including its temporal modeling and computational efficiency.

### G.1 TIME AWARENESS IN COGMIR

CogMir incorporates mechanisms to model belief and behavior evolution over time. While many social simulations (including some cognitive bias subsets in our paper) appear time-aware, they are fundamentally driven by events and memory. The perception of time emerges from the manipulation of these elements. By updating an agent's memory with prior context as new events unfold, a sense of temporal progression is created.

For example, in the Ben Franklin effect scenario, we construct a scenario where the LLM agent and human participants are in a job interview. As time progresses, the LLM agent's favorability towards the human interviewer changes. This time-dependent behavior arises because the agent performs a favor for the human interviewer during the interview. By manipulating the occurrence of events and updating the agent's memory, we can elicit evolving LLM Agents' behaviors over time, as demonstrated in the Ben Franklin effect scenario. This captures time-dependent phenomena through event and memory manipulation, effectively simulating the impact of time.

### G.2 COMPUTATIONAL EFFICIENCY OF COGMIR

Table 8 summarizes the API calls, simulated communication rounds, and token usage for our cognitive bias experiments.

API calls represent the calls per experimental question. Simulated communication rounds denote the actual inter-agent communications per experiment. Average token usage is averaged across all agents and questions per experiment.

The discrepancy between API calls and simulated rounds arises because not all participants are LLMs. Predefined prompts for human participants require no API calls. For example, in the Herd Effect,

Table 8: Cognitive Bias Experiments: API Calls, Communication Rounds, and Token Usage

| Cognitive Bias | API Calls | Simulated Communication Rounds | Average Token Usage |
|---|---|---|---|
| Herd Effect | 1 | $N + 1$ (N = participants) | $\sim 100 + N \times L$ |
| Authority Effect | 1 | 2 | $\sim 250$ |
| Ben Franklin Effect | 1 | 3 | $\sim 150$ |
| Confirmation Bias | 1 | 2 | $\sim 250$ |
| Halo Effect | 1 | 3 | $\sim 250$ |
| Rumor Chain Effect | $N$ | $N$ (N = agents) | $\sim 2 \times L \times N$ |
| Gambler's Fallacy | 1 | 1 | $\sim 150$ |

*Note:* $L$ represents average length of predefined messages, $N$ represents number of human participants/agents

human responses (e.g., all providing incorrect answers) are predefined, requiring only one API call for the LLM agent's response.

