# OpenReview forum: "Exploring Prosocial Irrationality for LLM Agents: A Social Cognition View"
_ICLR.cc/2025/Conference — ICLR 2025 Poster_

### Official Review · Reviewer_xMKu · 2024-10-27

**Soundness:** 3
**Presentation:** 2
**Contribution:** 3
**Rating:** 6
**Confidence:** 3

**Summary:**

This paper proposes to utilize the hallucinations of LLM-based agents to reflect and study human cognitive biases, which can further study irrational behaviors. It designs a new framework named CogMir, focusing on the study of social intelligence from hallucinations of LLMs. The results of extensive experiments show that CogMir can be aligned with real-world humans in terms of irrational and prosocial decision-making.

**Strengths:**

Pros:
- The idea of studying irrational behaviors from humans and LLM-based agents is very interesting. The pathway of reflecting human's irrational behaviors with LLM's hallucinations is fancy and interesting as well.
- The pipeline of CogMir is reasonable, which aligns the social simulations with real-world social experiments by outlining a standard workflow.
- The communication part models not only P2P settings but also the centralized (broadcast) settings, which is important for certain scenarios, such as social media trends.

**Weaknesses:**

Cons:
- How the LLM-based agents are designed? Are they just taking actions by prompting LLMs, rather than other modules like reasoning and memory? Could the authors elaborate on the architecture of the LLM agents? Specifically, are there any components beyond the base language model, such as reasoning modules or memory systems? A diagram or more detailed description of the agent architecture would be helpful.
- How do you deal with the factor of time? I think many social simulations are time-aware. Could the authors discuss how temporal aspects are handled in CogMir? For example, are there mechanisms to model the evolution of beliefs or behaviors over time in multi-turn interactions? Concrete examples of how time-dependent phenomena are captured would strengthen the paper.
- Could you please analyze the efficiency of CogMir? It would be helpful if the authors could provide an analysis of CogMir's computational efficiency. Specifically, how does the runtime scale with the number of agents or the complexity of scenarios? Are there any bottlenecks in the current implementation that could be addressed in future work?

**Questions:**

Please see the cons, and please point out if I have any misunderstandings.

---

> ### Author Response · Authors · 2024-11-20
>
> Thank you so much for your insightful feedback. We are excited that we resonated with each other in terms of ideas! We address your concerns below:
>
> **Response to Con 1** *How the LLM-based agents are designed? Are they just taking actions by prompting LLMs, rather than other modules like reasoning and memory? Could the authors elaborate on the architecture of the LLM agents? Specifically, are there any components beyond the base language model, such as reasoning modules or memory systems? A diagram or more detailed description of the agent architecture would be helpful.*
>
> The design of our LLM-based agent architecture aligns with existing *proof-of-concept* single-LLM agent systems [1,2]. Our primary contribution is the multi-LLM agents framework, CogMir, which investigates and interprets the irrational behaviors of LLM agents through cognitive biases. The architecture of the individual agent itself is not the focus of our work, so we did not provide explicit details on it in the main text.
>
> We appreciate your suggestion to include a diagram of the LLM agent architecture in the paper, as it will greatly help clarify and enhance our explanation. Below is a full description of the architecture of our individual LLM agent, which will be included in both text and diagrams in the appendix of the revised paper:
>
>
> ```
>                                +------------------------+
>                                |      Agent Profile     |
>                                +------------------------+
>                                            |
>                                            |
> +------------------------+      +----------v------------+      +-------------------+
> |    Memory Module       |<-----|         LLMs          |----->| Reasoning Module  |
> |   (History & Context)  |      |                       |      |                   |
> +------------------------+      +-----------------------+      +-------------------+
>             |                              |                             |
> +-----------------------+                  |                   +-------------------+
> |   Short-Term Memory   |                  |                   | Direct Inference  |
> +-----------------------+                  |                   +-------------------+
>                                 +----------v------------+      | Chain of Thought  |
>                                 |     Tools Module      |      +-------------------+
>                                 +-----------------------+
>                                            |
>                                 +-----------------------+
>                                 |  Point-to-Point Comm. |
>                                 +-----------------------+
>                                 |  Broadcast Comm.      |
>                                 +-----------------------+
> ```
> Our individual LLM Agent System includes four main components: the Memory Module, the Reasoning Module, the Agent Profile, and the Tools:
> 1. The `Memory Module` contains a Short-Term Memory sub-module to store recent context and history [3].
> 2. The `Reasoning Module` includes the Direct Inference and Chain of Thought Reasoning sub-modules [4], which will be activated based on the scenario. These two sub-modules are designed to mimic human reasoning patterns (e.g., inductive reasoning and deductive reasoning) [5,6].
> 3. The `Agent Profile` defines the agent's role and characteristics, where the actual role data is extracted from the `CogIdentity` dataset in the CogMir framework (see *Appendix C.4*).
> 4. The `Tools Module` includes communication protocols like Point-to-Point and Broadcast Communication, which allow agents to interact with other agents in the multi-LLM agents system.
>
> We give two examples to illustrate the system's operation.
> 1. In testing phenomena like the *herd effect*, we use the `CogIdentity` dataset to load the agent's role (e.g., a job applicant). When the experiment starts, predefined responses from other participants (e.g., all wrong answers) are loaded into the LLM’s short-term memory (`Memory Module`). The agent then gives its answer by direct inference, and we activate Chain of Thought reasoning to let the agent explain the rationale behind its answer (`Reasoning Module`).
> 2. The other example is the *rumor chain effect*, we first define the scenario and give the initial message to pass. Every agent will use the `Point-to-Point` communication tool to pass the messages one by one （`Tools Module`）. Here the memory module is not activated as the agents are not allowed to know the history of the message (`Memory Module`). Since we want to simulate how rumors are spread in real life, the agents will only use Direct Inference to pass the message rather than take thoughtful consideration (`Reasoning Module`).

---

> > ### Author Response · Authors · 2024-11-20
> > **Continue Response to Con1**
> >
> > Our framework is designed to be open-ended and extensible. The broader architecture for individual LLM Agent Systems includes components like a long-term memory system for persistent knowledge storage and retrieval [7], and more complex reasoning strategies (e.g., self-reflection [2]) to emulate human-like deliberation. These advanced components, though not utilized in the present study, are part of our ongoing work.
> >
> > [1] Park, Joon Sung, Joseph O'Brien, Carrie Jun Cai, Meredith Ringel Morris, Percy Liang, and Michael S. Bernstein. "Generative agents: Interactive simulacra of human behavior." UIST, 2023.
> >
> > [2] Shinn, Noah, Federico Cassano, Ashwin Gopinath, Karthik Narasimhan, and Shunyu Yao. "Reflexion: Language agents with verbal reinforcement learning." NeurIPS, 2024.
> >
> > [3] Qingxiu Dong, Lei Li, Damai Dai, Ce Zheng, Jingyuan Ma, Rui Li, Heming Xia, Jingjing Xu, Zhiyong Wu, Baobao Chang, Xu Sun, Lei Li, and Zhifang Sui. "A Survey on In-context Learning." EMNLP, 2024.
> >
> > [4] Wei, Jason, Xuezhi Wang, Dale Schuurmans, Maarten Bosma, Fei Xia, Ed Chi, Quoc V. Le, and Denny Zhou. "Chain-of-thought prompting elicits reasoning in large language models." NeurIPS, 2022.
> >
> > [5] Heit, Evan. "Properties of inductive reasoning." Psychonomic bulletin & review, 2000.
> >
> > [6] Johnson-Laird, Philip N. "Deductive reasoning." Annual Review of psychology, 1999.
> >
> > [7] Zhang, Kai, Yangyang Kang, Fubang Zhao, and Xiaozhong Liu. "LLM-based Medical Assistant Personalization with Short- and Long-Term Memory Coordination." NAACL,2024.

---

> > > ### Author Response · Authors · 2024-11-20
> > >
> > > **Response to Con 2** *How do you deal with the factor of time? I think many social simulations are time-aware. Could the authors discuss how temporal aspects are handled in CogMir? For example, are there mechanisms to model the evolution of beliefs or behaviors over time in multi-turn interactions? Concrete examples of how time-dependent phenomena are captured would strengthen the paper.*
> > >
> > > Thanks for raising the question regarding how CogMir addresses temporal aspects.
> > >
> > > CogMir does incorporate mechanisms to model belief and behavior evolution over time. It’s worth noting that while many social simulations (including some cognitive bias subsets in our paper) are time-aware, they are not inherently "time-aware." Instead, they are often "event-aware" or "memory-aware."
> > >
> > > Therefore, the effect of time can be captured through event and memory manipulation. When different events occur, we correspondingly update the agent's memory and provide the agents with the previous context, which gives agents a sense of time progression.
> > >
> > > For example, in the Ben Franklin effect scenario, we construct a scenario where the LLM agent and human participants are in a job interview. As time progresses, we observe that the LLM agent's favorability towards the human interviewer changes. This time-dependent behavior is because the agent does a favor for the human interviewer during the interview. By manipulating the occurrence of events and updating the agent's memory, we can elicit evolving LLM behaviors over time, as shown in the Ben Franklin effect scenario. This captures time-dependent phenomena through event and memory manipulation, effectively simulating the impact of time. A brief discussion will be included in the *Appendix*.

---

> > > > ### Author Response · Authors · 2024-11-20
> > > >
> > > > **Response to Con 3** *Could you please analyze the efficiency of CogMir? It would be helpful if the authors could provide an analysis of CogMir's computational efficiency. Specifically, how does the runtime scale with the number of agents or the complexity of scenarios? Are there any bottlenecks in the current implementation that could be addressed in future work?*
> > > >
> > > > **1. Suggestion** *Analyze the efficiency of CogMir*
> > > >
> > > > We appreciate your suggestions on analyzing the efficiency of CogMir; we will include the following table and analysis in the *Appendix*:
> > > >
> > > > This table presents the API calls and simulated communication rounds for the experiments conducted to test LLM cognitive biases.
> > > >
> > > > | Cognitive Bias Experiment | API Calls | Simulated Communication Rounds       | Average Token usage
> > > > | ------------------------- | --------- | ------------------------------------ |---------------------
> > > > | Herd Effect               | 1         | N+1 (N = number human participants)  | ~100 + N * L (L = average length of human response)
> > > > | Authority Effect          | 1         | 2                                    | ~250
> > > > | Ben Franklin Effect       | 1         | 3                                    | ~150
> > > > | Confirmation Bias         | 1         | 2                                    | ~250
> > > > | Halo Effect               | 1         | 3                                    | ~250
> > > > | Rumor Chain Effect        | N         | N (N = total number of agents)       | ~2 * L * N (L = average length of message, N = number of agents)
> > > > | Gambler's Fallacy         | 1         | 1                                    |~150
> > > >
> > > > Note that the API Calls are the number of APIs used for one experimental question. Simulated Communication Rounds are the number of actual number f inter-agent communication in one experiment. Average Token usage is the number of tokens averaged over all agents and all questions for on experiment.
> > > >
> > > > The number of API calls is fewer than the number of simulated communication rounds because not all participants are LLM agents; thus, some communication prompts are predefined and do not require additional API calls. For example, in the Herd Effect experiment, the human participants' responses are predefined (e.g., all giving wrong answers), and only the LLM agent's response requires an API call.
> > > >
> > > >
> > > > **2. Questions** *Current Bottlenecks in the current implementation that could be addressed in future work*
> > > >
> > > > As to the bottlenecks in our current implementation, we think we have a bottleneck in modeling a more realistic "rumor chain effect" in real-life scenarios. Although we have modeled the *rumor chain effect* efficiently using our *Series* (Point-to-point) protocols, we haven't considered the impact of random noise when passing the messages. For example, in real-life scenarios when rumor spreads, apart from agents' distortion of the message, there could be other irrelevant random effects such as misheard or misspellings which are not caused by cognitive distortion.
> > > > With this in mind, we plan to extend our system to incorporate a certain level of random perturbation during the message-passing process. Since LLM employs a decoder-only transformer structure, its output is an embedded vector which will then be converted to text. With a predefined noise level (according to different real-life scenarios), we can perturb the vector before converting it to text, simulating the irrelevant random noise in real-life message passing. This will make the modeling of *rumor chain effect* more realistic in our future work, giving a more complete picture.

---

> > > > > ### Comment · Reviewer_xMKu · 2024-11-22
> > > > >
> > > > > Thanks for the detailed rebuttal by the authors. The rebuttal has addressed my concerns. I would like to maintain my score of 6, and I prefer to accept this paper.

---

> ### Author Response · Authors · 2024-11-23
>
> Dear Reviewer xMKu,
>
> Thank you for your feedback and for taking the time to review our rebuttal.
>
> We are glad to hear that your concerns have been addressed, and we truly appreciate your continued support.
>
> Wishing you all the best！

---

> > ### Author Response · Authors · 2024-11-26
> > **Revised paper has been uploaded**
> >
> > Dear Reviewer xMKu,
> >
> > We have uploaded the revised version of our paper. Thanks again for taking the time to read our response and engaging in the discussion.
> >
> > All the best!

---

### Official Review · Reviewer_mwaV · 2024-10-28

**Soundness:** 3
**Presentation:** 3
**Contribution:** 2
**Rating:** 5
**Confidence:** 2

**Summary:**

The authors explore using hallucinations from LLMs to mirror human cognitive biases. They leverage the hallucination property of LLMs to assess and enhance LLM Agents' social intelligence through cognitive biases. Their proposed framework, CogMir, indicates that LLM Agents have pro-social behavior in irrational decision making.

**Strengths:**

- systematic construction of framework to assess and interpret social intelligence of agents
- The paper throughout is engaging, and methodical, self-contained for readers that are not familiar with the topic. Overall very well written.

**Weaknesses:**

- missing distributional metrics, since the temperature is set to 1 for all models, might be hard to reproduce the results. Also the conclusions drawn from these results might be invalid due to variance.
- The LLM Agent architectures/prompts used were not presented in the paper.

**Questions:**

- The prompts used for the agents are not presented. How are the prompts constructed and "tuned"?

---

> ### Author Response · Authors · 2024-11-20
>
> Thank you very much for your feedback! We would like to clarify some misunderstandings regarding the weaknesses and address your concerns below:
>
> **Clarify for Weakness 1**: *missing distributional metrics, since the temperature is set to 1 for all models, might be hard to reproduce the results. Also, the conclusions drawn from these results might be invalid due to variance.*
>
> We want to address the misunderstanding regarding the temperature configuration.
>
> Setting the temperature to 1 for all models does not affect the reproducibility of the results in our task. As explained in the *Experiments & Discussion* section, the temperature does not influence our conclusions because we conducted repeated experiments for each case: each evaluation of the same inquiry was performed 10 times (with over 1000 queries per case). The large number of repeated experiments allows us to have strong `statistical power` to theoretically confirm the reproducibility and validity of our findings. This approach is consistent with previous social science studies where the focus is on the overall effect size and statistical significance rather than individual sample variations. We calculated `statistical significance` for each bias and found the results robust enough to conclude [1].
>
> Here we present one example showing how we get the `statistical significance` to validate our solution:
>
> **Example:** We carried out 10 * 100 repeated experiments for the Ben Franklin effect for each model. This could be considered as testing 10 agents with 100 questions.
> Let $B_i$ be the bias rate for agent $i$ across $n=100$ questions. With $m=10$ agents, the null hypothesis ($H_0$) is that `Mixtral-8x7b` does not exhibit the Ben Franklin effect ($\mu = E[B_i] \leq 0.5$). Here we use $n=100$ questions which provide strong `statistical power`. We use a standard Z-test to test the null hypothesis. Observing $\bar{B} = 0.66$ with a variance of at most $0.20$ ($\sigma \leq 0.20$), a one-tailed Z-test yields:
>
> $Z = \frac{\bar{B} - \mu_0}{\sigma/\sqrt{m}} = \frac{0.66 - 0.5}{0.20/\sqrt{10}} \approx 2.53$
>
> This results in a p-value of $P(Z > 2.53) \approx 0.0057$. This indicates that the probability of observing a bias rate of 0.66 or higher under the null hypothesis is less than 0.01. Thus, we have strong statistical significance to reject the null hypothesis and conclude that `Mixtral-8x7b` exhibits the Ben Franklin effect.
>
> Moreover, our task in this work is to analyze and interpret the systematic irrational behaviors of LLM agents. Regardless of the temperature setting, the overall outcome remains consistent. Instead of assessing cognitive bias based on the similarity of specific words (which is influenced by temperature), *we focus on the overall meaning conveyed by the text (which remains independent of temperature) *.
>
> Additionally, this temperature configuration is commonly used in previous studies investigating the social intelligence of LLM agents. Following [2], we set the agents' temperature to 1 to promote diversity in their responses, while the evaluator's temperature is set to 0 to ensure stability in the evaluation process.
>
> [1] Hanushek, Eric A., and John E. Jackson. "Statistical methods for social scientists." Academic Press, 2013.
>
> [2] Zhou*, Xuhui, Zhu*, Hao, Mathur, Leena, Zhang, Ruohong, Qi, Zhengyang, Yu, Haofei, Morency, Louis-Philippe, Bisk, Yonatan, Fried, Daniel, Neubig, Graham, and Sap, Maarten. "SOTOPIA: Interactive Evaluation for Social Intelligence in Language Agents." ICLR, 2024.

---

> > ### Author Response · Authors · 2024-11-20
> >
> > **Clarify and Response on Weakness 2**:*The LLM Agent architectures/prompts used were not presented in the paper.*
> >
> > **1.Clarify** : *Agent prompts for all cases were included in the main text and appendix.*
> >
> > In our paper, we presented sample agent prompt templates in the main text and appendix. Due to space limitations, in the *Experiments & Discussion* section in the main text, we only provide necessary simplified prompt templates. The full sample prompt templates, along with detailed instructions on how to use them for each case, can be found in *Appendix Section D*.
> >
> > **2.Response** : *Detailed Agent Architecture will be added to the appendix.*
> >
> > The LLM agent architecture in our work aligns with existing *proof-of-concept* single-LLM agent systems [3,4]. Our primary contribution is the multi-LLM agents framework, CogMir, which investigates and interprets the irrational behaviors of LLM agents through cognitive biases. The architecture of the individual agent itself is not the focus of our work, so we did not provided explicit details on it in the main text.
> >
> > Below is a full description of the architecture of our individual LLM agent, which will be included in both text and diagrams in the appendix of the revised paper:
> >
> > ```
> >                                +------------------------+
> >                                |      Agent Profile     |
> >                                +------------------------+
> >                                            |
> >                                            |
> > +------------------------+      +----------v------------+      +-------------------+
> > |    Memory Module       |<-----|         LLMs          |----->| Reasoning Module  |
> > |   (History & Context)  |      |                       |      |                   |
> > +------------------------+      +-----------------------+      +-------------------+
> >             |                              |                             |
> > +-----------------------+                  |                   +-------------------+
> > |   Short-Term Memory   |                  |                   | Direct Inference  |
> > +-----------------------+                  |                   +-------------------+
> >                                 +----------v------------+      | Chain of Thought  |
> >                                 |     Tools Module      |      +-------------------+
> >                                 +-----------------------+
> >                                            |
> >                                 +-----------------------+
> >                                 |  Point-to-Point Comm. |
> >                                 +-----------------------+
> >                                 |  Broadcast Comm.      |
> >                                 +-----------------------+
> > ```
> > Our individual LLM Agent System includes four main components: the Memory Module, the Reasoning Module, the Agent Profile, and the Tools:
> > 1. The `Memory Module` contains a Short-Term Memory sub-module to store recent context and history [5].
> > 2. The `Reasoning Module` includes the Direct Inference and Chain of Thought Reasoning sub-modules [6], which will be activated based on the scenario. These two sub-modules are designed to mimic human reasoning patterns (e.g., inductive reasoning and deductive reasoning) [7,8].
> > 3. The `Agent Profile` defines the agent's role and characteristics, where the actual role data is extracted from the `CogIdentity` dataset in the CogMir framework (see *Appendix C.4*).
> > 4. The `Tools Module` includes communication protocols like Point-to-Point and Broadcast Communication, which allow agents to interact with other agents in the multi-LLM agents system.
> >
> > We give two examples to illustrate the system's operation.
> > 1. In testing phenomena like the *herd effect*, we use the `CogIdentity` dataset to load the agent's role (e.g., a job applicant). When the experiment starts, predefined responses from other participants (e.g., all wrong answers) are loaded into the LLM’s short-term memory (`Memory Module`). The agent then gives its response by direct inference, and we activate Chain of Thought reasoning to let the agent explain the rationale behind its response (`Reasoning Module`).
> > 2. The other example is the *rumor chain effect*, we first define the scenario and give the initial message to pass. Every agent will use the `Point-to-Point` communication tool to pass the messages one by one （`Tools Module`）. Here the memory module is not activated as the agents are not allowed to know the history of the message (`Memory Module`). Since we want to simulate how rumors are spread in real life, the agents will only use Direct Inference to pass the message rather than take thoughtful reasoning (`Reasoning Module`).

---

> > > ### Author Response · Authors · 2024-11-20
> > > **Continue to Clarify and Response on Weakness 2**
> > >
> > > Our framework is designed to be open-ended and extensible. The broader architecture for individual LLM Agent Systems includes components like a long-term memory module for persistent knowledge storage and retrieval [9], and more complex reasoning strategies (e.g., self-reflection [4]) to emulate human-like deliberation. These advanced components, though not utilized in the present study, are part of our ongoing work.
> > >
> > >
> > > [3] Park, Joon Sung, Joseph O'Brien, Carrie Jun Cai, Meredith Ringel Morris, Percy Liang, and Michael S. Bernstein. "Generative agents: Interactive simulacra of human behavior." UIST, 2023.
> > >
> > > [4] Shinn, Noah, Federico Cassano, Ashwin Gopinath, Karthik Narasimhan, and Shunyu Yao. "Reflexion: Language agents with verbal reinforcement learning." NeurIPS, 2024.
> > >
> > > [5] Qingxiu Dong, Lei Li, Damai Dai, Ce Zheng, Jingyuan Ma, Rui Li, Heming Xia, Jingjing Xu, Zhiyong Wu, Baobao Chang, Xu Sun, Lei Li, and Zhifang Sui. "A Survey on In-context Learning." EMNLP, 2024.
> > >
> > > [6] Wei, Jason, Xuezhi Wang, Dale Schuurmans, Maarten Bosma, Fei Xia, Ed Chi, Quoc V. Le, and Denny Zhou. "Chain-of-thought prompting elicits reasoning in large language models." NeurIPS, 2022.
> > >
> > > [7] Heit, Evan. "Properties of inductive reasoning." Psychonomic bulletin & review, 2000.
> > >
> > > [8] Johnson-Laird, Philip N. "Deductive reasoning." Annual Review of psychology, 1999.
> > >
> > > [9] Zhang, Kai, Yangyang Kang, Fubang Zhao, and Xiaozhong Liu. "LLM-based Medical Assistant Personalization with Short- and Long-Term Memory Coordination." NAACL,2024.

---

> > > > ### Author Response · Authors · 2024-11-20
> > > >
> > > > **Clarify & Answer Question**: *The prompts used for the agents are not presented. How are the prompts constructed and "tuned"?*
> > > >
> > > >
> > > > **1. Clarify** : As discussed in our response to Weakness 2, we provided sample prompt templates and instructions on “tuned” in *Appendix Section D*.
> > > >
> > > > **2. Answer** : Below, we offer a more detailed explanation of how we construct and tune the prompts for the agents.
> > > >
> > > >
> > > > Our prompt templates consist of two parts: *Fixed Context* and *Scene-Adjustable Features*. Prompts are customized ("tuned") by populating *Scene-Adjustable Features* with specific instances:
> > > >
> > > > 1. *Fixed Context*: The *Fixed Context* is designed based on well-established human experimental paradigms to simulate various social experiments, ensuring both validity and relevance. Essential *Scene-Adjustable Features*—such as scenarios, actions, identities—are embedded as placeholders (in square brackets) within the *Fixed Context*.
> > > >
> > > > 2. *Scene-Adjustable Features*: We use advanced LLMs, such as GPT-4, to generate instances of different *Scene-Adjustable Features* (with the exception of 'MCQ', which contains factual information sourced from open databases such as Wikipedia). These generated instances are compiled into datasets that populate the prompts with diverse and realistic examples. The datasets (e.g., `CogScene`, `CogIdentity`, etc.) undergo manual review to ensure quality, clarity, and diversity (see *Appendix Section C*). Placeholders in the prompt template are filled with instances from the datasets to create specific, diverse prompts.
> > > >
> > > > Aligned with the structure of CogMir (*Section 3.4* in the main text), our prompts are categorized into *Simulation* and *Survey* types.
> > > > Here are some examples of how we construct and tune the prompts for specific biases for both *Simulation* and *Survey*:
> > > >
> > > > 1. *Simulation* Example-Authority Effect Simulation Prompt:
> > > >
> > > >      Simulation prompt template for the Authority Effect (from *Appendix D.4*):
> > > >      ```
> > > >      Round #1: As a [INFERIOR IDENTITY: student], you are [SCENARIO: in a classroom taking a math course]. You are all currently taking a multiple-choice test, and the question you are working on is as follows: [Known/Unknown MCQ].
> > > >      Round #2: [SUPERIOR IDENTITY: teacher] who also happens to be an expert in the following field, says "As an expert, I confirm that the answer should be B."
> > > >      Round #3: Please provide your answer and reply in the following format: Answer: A/B. Explanation: ***.
> > > >      ```
> > > >      For specific prompts, we fill in the placeholders with the corresponding instances from the `CogScene` `CogIdentity`, and `MCQ` datasets. This is an actual prompt used in OpenAI API:
> > > >      ```python
> > > >      {
> > > >           "system": "As a student, you are in a classroom taking a math course. You are all currently taking a multiple-choice test, and the question you are working on is as follows: What is the color of the apple? A: Red, B: Blue.",
> > > >           "system": "Your teacher who also happens to be an expert in the following field, says"
> > > >           "user": "As an expert, I confirm that the answer should be B.",
> > > >           "system": " Please provide your answer and reply in the following format: Answer: A/B. Explanation: ***."
> > > >      }
> > > >      ```
> > > >
> > > >
> > > > 2. *Survey* Example-Authority Effect Survey Prompt:
> > > >
> > > >      Simulation prompt template for the Authority Effect (from *Appendix D.4*):
> > > >
> > > >      ```
> > > >      As a [INFERIOR IDENTITY: student], you are [SCENARIO: in a classroom taking a math course]. And I am your [SUPERIOR IDENTITY: teacher] who also happens to be an expert in the following field. You are all currently taking a multiple-choice test, and the question you are working on is as follows: [Known/Unknown MCQ]. As an expert, I confirm that the answer should be B. Please provide your answer and reply in the following format: Answer: A/B. Explanation: ***
> > > >      ```
> > > >      For specific prompts, we fill in the placeholders with the corresponding values from the `CogScene` `CogIdentity`, and `MCQ` datasets. This is an actual prompt used in OpenAI API:
> > > >      ```python
> > > >      {
> > > >           "system": "As a student, you are in a classroom taking a math course. And I am your teacher who also happens to be an expert in the following field. You are all currently taking a multiple-choice test, and the question you are working on is as follows: What is the color of the apple? A: Red, B: Blue. As an expert, I confirm that the answer should be B. Please provide your answer and reply in the following format: Answer: A/B. Explanation: ***",
> > > >      }
> > > >      ```
> > > >
> > > > Prompts for other biases follow similar structures tailored to their respective experimental settings. For more detailed examples and templates, please refer to  *Appendix D* for more details.

---

> > > > > ### Author Response · Authors · 2024-11-25
> > > > >
> > > > > Dear Reviewer mwaV,
> > > > >
> > > > > Thank you again for taking the time to review our paper. Does our rebuttal address your concerns?
> > > > >
> > > > > Your feedback and support are very important to us.
> > > > >
> > > > > Best Regards,
> > > > >
> > > > > Authors of Submission 5862 Exploring Prosocial Irrationality for LLM Agents: A Social Cognition View

---

> > > > > > ### Author Response · Authors · 2024-11-26
> > > > > > **Revised paper has been uploaded**
> > > > > >
> > > > > > Dear Reviewer mwaV,
> > > > > >
> > > > > > We have uploaded the revised version of our paper.
> > > > > >
> > > > > > Thank you again for taking the time to review our paper. Does our rebuttal address your concerns?
> > > > > >
> > > > > > Your feedback and support are very important to us.
> > > > > >
> > > > > > Best Regards,
> > > > > >
> > > > > > Authors of Submission 5862 Exploring Prosocial Irrationality for LLM Agents: A Social Cognition View

---

> > ### Author Response · Authors · 2024-12-02
> >
> > Dear Reviewer mwaV
> >
> > The deadline for reviewer feedback is approaching (December 2nd), only one day left. Do our rebuttal and the revised paper address your concerns?
> >
> > Thank you again for your thorough review. Your support and feedback are invaluable to us.
> >
> > Best regards,
> >
> > Authors of Submission 5862 Exploring Prosocial Irrationality for LLM Agents: A Social Cognition View

---

> ### Author Response · Authors · 2024-12-04
>
> Dear Reviewer mwaV,
>
> We are still actively awaiting your further engagement and hope our rebuttal and the revised paper address your concerns.
>
> Thank you again for your time and valuable feedback.
>
> Best regards,
>
> Authors of Submission 5862 Exploring Prosocial Irrationality for LLM Agents: A Social Cognition View

---

### Official Review · Reviewer_Ko1b · 2024-11-05

**Soundness:** 2
**Presentation:** 2
**Contribution:** 2
**Rating:** 6
**Confidence:** 4

**Summary:**

This submission proposes CogMir, a novel framework for evaluating the "social intelligence" of Large Language Model (LLM) Agents by assessing their susceptibility to cognitive biases traditionally studied in social science.  The framework leverages the "hallucination" properties of LLMs, positing that these deviations from factual accuracy can mirror human cognitive biases, thus offering a window into the development of irrational social intelligence in agents. CogMir integrates established social science experiments, adapting them into human-LLM and multi-agent interaction scenarios, along with multiple evaluation methods (human, LLM, dataset-based, and using state-of-the-art discriminators). The authors conduct experiments on a subset of cognitive biases, demonstrating consistency between LLM agents and human responses in prosocial bias contexts, while highlighting differences in non-prosocial and certainty/uncertainty situations.

**Strengths:**

Novel Approach: The paper presents a unique and intriguing approach to evaluating LLM agent social intelligence by connecting it to the well-established field of human cognitive biases. This bridges a gap in current LLM Agent research that often focuses on black-box testing.

Well-Defined Framework: CogMir is described as a modular and extensible framework, encompassing various components for experimental settings, interaction modes, cognitive bias subsets, and evaluation metrics. This provides a structured and replicable methodology for future research in this area.

Transparency and Reproducibility: The paper provides details about the datasets used, evaluation methods employed, and experimental setup, enhancing the potential for reproducibility and allowing others to build upon this work.  The inclusion of an appendix with further details is also commendable.

**Weaknesses:**

Lack of a Robust Definition of Social Intelligence: The paper relies on a superficial association between cognitive biases and "social intelligence" without a clear theoretical grounding, hence the specifics of social intelligence being measured remain ambiguous. It's difficult to contextualize what the findings actually suggest without such grounding. It may be beneficial to have a more rigorous and operationalized definitions.

Insufficient Engagement with Social and Cognitive Theory: The paper's theoretical framework is underdeveloped.  While it mentions evolutionary psychology, it lacks a deep engagement with broader social and cognitive theories. Concepts like bounded rationality and cognitive bounds, which recognize inherent limitations on human cognitive processing and decision-making, are more relevant to understanding the role of biases in shaping social behavior. For example, we have been exploring computational models of bounded rationality in our field, it would make a lot of sense to compare notes with. Irrationality requires a clear definition of what it means to be rational, which isn't present in this submission. Incorporating such theoretical perspectives would provide a stronger foundation for interpreting the findings.

Limited Scope of Cognitive Biases and Social Scenarios: The paper's focus on a small subset of cognitive biases risks an incomplete picture of LLM agent social intelligence.  Similarly, the simulated social scenarios, while helpful for initial exploration, lack the complexity of real-world social interactions.

The promised connection between hallucination and cognitive biases are missing.  It seems like there are certain types of human cognitive biases that are particularly easily translatable with "hallucination". I wonder if it's worth narrowing down and finding a crisp connection. Also framing "hallucination" as a potential source of social intelligence needs careful consideration. The paper needs to explicitly address the potential negative impacts of inaccurate and biased outputs in real-world settings. The ethical implications of replicating human cognitive biases, especially harmful ones, in LLMs require a more thorough discussion.

Methodology Limitations:  While the paper strives for transparency and replicability, further validation of the evaluation metrics and exploration of alternative assessment methods would strengthen the robustness of the findings.


----
I sincerely apologize for engaging with the submission just now after the extended deadline.

I thank the authors for clarifying some of the issues raised. While the authors have provided additional citations and their interpretations - I would have liked a stronger grounding in cognitive modeling and cognitive theory literature. Specifically not through authors' own interpretations, but through experiments or logical reduction to theories that are particularly important in the assumptions authors make. I remain concerned on this end.

I was able to resolve some misunderstandings I had by author responses. While I still think the claimed contributions need additional validation, I will remain reserved. There's greater value in replicating social experiments with LLM agents in a systematic manner, and it does have a path forward if the authors spend more space for explaining their agent design/architecture to make tighter connections to the results.

**Questions:**

How can the CogMir framework be used to understand more complex social interactions, such as negotiation, persuasion, and deception?

What are the implications of designing LLM Agents with cognitive biases? How does the performance of LLMs with different architectures and training datasets vary in their susceptibility to cognitive biases?

Can the insights from CogMir be used to improve LLM Agent training and design, leading to more socially intelligent and responsible AI systems? Does mirroring human biases in LLMs might simply perpetuate and amplify existing societal biases?

Authors have selected seven "state-of-the-art" models in line 289, yet the models aren't state of the art. Any further explanations?

---

I thank the authors for answering my questions. I'm still unsure if we readers can grasp how sensitive the results are when experimented with other newer models. Some validation of generalizability across models would be beneficial. Though I think it's a peripheral issue to the main constribution - I would suggest only to rephrase (perhaps delete?) mentions of "SOTA" to avoid confusions.

---

> ### Author Response · Authors · 2024-11-21
>
> Thank you for your feedback and suggestions. We would like to clarify some misunderstandings and address your concerns below：
>
>
> **Clarify for Weakness 1**: *Lack of a Robust Definition of Social Intelligence: The paper relies on a superficial association between cognitive biases and "social intelligence" without a clear theoretical grounding, hence the specifics of social intelligence being measured remain ambiguous. It's difficult to contextualize what the findings suggest without such grounding. It may be beneficial to have more rigorous and operationalized definitions.*
>
> Our paper has the theoretical foundation for the link between cognitive biases and social intelligence, as mentioned in *Section 1 INTRODUCTION*. Here, we provide a more detailed explanation of how we define social intelligence and establish the theoretical grounding for our work. We will include the following definitions and explanations in the *Appendix* in revised paper.
>
> **Defining Social Intelligence**
>
> Following [1], we define social intelligence as the ability to get along with others, encompassing interpersonal skills related to cooperation, communication, and competition within personal and social value systems. This definition emphasizes social intelligence's **interactive** and **communication** nature.
>
> **Theoretical Foundation Linking Cognitive Biases and Social Intelligence**
>
> The theoretical foundation for the link between cognitive biases and social intelligence is established through insights drawn from evolutionary biology, game theory, and psychology. Research using computational models and game theory indicates that **cognitive biases** can foster cooperation and communication, the fundamental aspects of **social intelligence** [2,3,4]. Furthermore, some **cognitive biases** were found to improve large-scale decision-making, reinforcing their connection to **social intelligence** [5,6].
>
> These cross-disciplinary findings provide compelling evidence for the deep theoretical link between cognitive biases and social intelligence. This theoretical grounding established the validity of our work in finding LLM Agents' social intelligence from their demonstrated “cognitive biases” in their irrational behaviors.
>
>
> [1] Kihlstrom, John F., and Nancy Cantor. "Social intelligence." Handbook of intelligence 2 (2000): 359-379.
>
> [2] Vogrin, Michael, Guilherme Wood, and Thomas Schmickl. "Confirmation bias as a mechanism to focus attention enhances signal detection." Journal of Artificial Societies and Social Simulation 26, no. 1 (2023).
>
> [3] Yang, Han-Xin, Wen-Xu Wang, Zhi-Xi Wu, Ying-Cheng Lai, and Bing-Hong Wang. "Diversity-optimized cooperation on complex networks." Physical Review E—Statistical, Nonlinear, and Soft Matter Physics 79, no. 5 (2009): 056107.
>
> [4] Quan, Ji, Haoze Li, and Xianjia Wang. "Cooperation dynamics in public goods games with evolving cognitive bias." Management System Engineering 2, no. 1 (2023): 15.
>
> [5] Mishra, Sandeep. "Decision-making under risk: Integrating perspectives from biology, economics, and psychology." Personality and Social Psychology Review 18, no. 3 (2014): 280-307.
>
> [6] Johnson, Dominic DP, Daniel T. Blumstein, James H. Fowler, and Martie G. Haselton. "The evolution of error: Error management, cognitive constraints, and adaptive decision-making biases." Trends in ecology & evolution 28, no. 8 (2013): 474-481.

---

> > ### Author Response · Authors · 2024-11-21
> >
> > **Clarify for Weakness 3**: *Limited Scope of Cognitive Biases and Social Scenarios: The paper's focus on a small subset of cognitive biases risks an incomplete picture of LLM agent social intelligence. Similarly, the simulated social scenarios, while helpful for initial exploration, lack the complexity of real-world social interactions.*
> >
> > **1. Clarify for Limited Scope of Cognitive Biases:**
> >
> > In our paper, our chosen subset is representative and diverse enough to provide a comprehensive scope of how LLM agents exhibit social intelligence through their irrational behaviors.
> >
> > Since there are hundreds of cognitive biases, it is not feasible to test all of them. Therefore, we take a comprehensive literature review [8,9,10,11] to select a subset of cognitive biases that are most representative and of the highest significance in the related field.
> >
> > To make our subset diverse, we followed the principles in [8], where the author points out it is the unbounded rationality [12] that leads agents to rely on "Effort-Reduction" principles. These principles are the foundation of heuristics, which in turn leads to cognitive biases. The author proposes a framework for studying heuristics, which includes five rules. Following these rules, we selected our tested cognitive biases to cover a wide range of causes of heuristics to provide a diverse set:
> >
> > |        Effort-Reduction Principles        |                Biases                |
> > | :---------------------------------------: | :----------------------------------: |
> > |           Examining fewer cues            |             Herd Effect              |
> > |     Retrieving and storing cue values     |          Rumor Chain Effect          |
> > | Simplifying weighting principles for cues |    Halo Effect, Authority Effect     |
> > |       Integrating less information        | Confirmation Bias, Gambler's Fallacy |
> > |       Examining fewer alternatives        |         Ben Franklin Effect          |
> >
> > In this way, we ensure that our selected biases cover a wide range of cognitive processes that can lead to irrational decision-making, thus providing a comprehensive understanding of LLM agents' social intelligence.
> >
> >
> > [8] Shah, Anuj K., and Daniel M. Oppenheimer. "Heuristics made easy: an effort-reduction framework." Psychological Bulletin 134, no. 2 (2008): 207.
> >
> > [9] Haselton, Martie G., Daniel Nettle, and Paul W. Andrews. "The evolution of cognitive bias." The handbook of evolutionary psychology (2015): 724-746.
> >
> > [10] Hilbert, Martin. "Toward a synthesis of cognitive biases: how noisy information processing can bias human decision making." Psychological Bulletin 138, no. 2 (2012): 211.
> >
> > [11] Thomas, Oliver. "Two decades of cognitive bias research in entrepreneurship: what do we know and where do we go from here?." Management Review Quarterly 68, no. 2 (2018): 107-143.
> >
> > [12] Gigerenzer, Gerd, and Reinhard Selten. "Rethinking rationality." (2001).
> >
> >
> >
> > **2. Clarify for Limited Scope of Social Scenarios**
> >
> > Please note that CogMir is not designed to simulate "social scenarios" but to mirror established human social science experiments. In our mirrored experiments, the tested subjects are replaced with LLM Agents from the original human participants. This allows us to use well-established social science experiments in investigating human cognitive biases to observe, explain, and explore LLM Agents' irrational behaviors.
> >
> > As we explained in our paper, we have successfully mirrored human cognitive biases by hallucination in LLM Agents and systematically explained the irrational behaviors of LLM Agents through rigorous social science methodologies. The results we obtained are generally applicable the same as those social science results. Therefore, our finding is generalizable to real-world AI applications involving diverse human inputs.

---

> > > ### Author Response · Authors · 2024-11-21
> > >
> > > **Clarify for Weakness 4**: *The promised connection between hallucination and cognitive biases is missing. It seems like certain types of human cognitive biases are particularly easily translatable with "hallucination." I wonder if it's worth narrowing down and finding a crisp connection. Also framing "hallucination" as a potential source of social intelligence needs careful consideration. The paper needs to explicitly address the potential negative impacts of inaccurate and biased outputs in real-world settings. The ethical implications of replicating human cognitive biases, especially harmful ones, in LLMs, require a more thorough discussion.*
> > >
> > > **1. Clarify the concern on the connection between hallucination and cognitive biases**
> > >
> > > In our *Section 1 INTRODUCTION*, we show that **systematic hallucination** is closely related to human **cognitive biases** [14]. Here, we provide a more detailed explanation for this argument.
> > >
> > > In evolutionary psychology [9], heuristics are deemed as one major cause of cognitive bias aligned with the theory of bounded rationality [13].
> > > Heuristics in human cognition are defined as mental shortcuts that can help humans make faster decisions, which typically involve considering the most relevant or familiar aspects of a problem [18]. Similarly, several recent studies have noticed that LLM generally tends to give answers through "pattern matching," which is also a heuristic method of finding familiar structures and templates [15,16,17] and tends to cause systematic hallucination. With this theoretical evidence on the similarity between LLM and human decision-making patterns, we argue that systemic hallucination in LLM Agents can mirror human cognitive bias.
> > >
> > > **2. Clarify the concern on potential ethical implications**
> > >
> > > Our framework successfully mirrored human cognitive biases by hallucination in LLM Agents and systematically explained the irrational behaviors of LLM Agents. Therefore, our study presents a powerful tool for detecting LLM's biased behavior in real-world settings. Replicating cognitive bias in humans doesn't contribute to ethical issues; instead, it helps us better prevent them and develop more trustworthy and reliable LLM Agent applications in the real world.
> > >
> > >
> > > [13] Simon, Herbert A. "Bounded rationality." Utility and probability (1990): 15-18.
> > >
> > > [14] Ji, Ziwei, Nayeon Lee, Rita Frieske, Tiezheng Yu, Dan Su, Yan Xu, Etsuko Ishii, Ye Jin Bang, Andrea Madotto, and Pascale Fung. "Survey of hallucination in natural language generation." ACM Computing Surveys 55, no. 12 (2023): 1-38.
> > >
> > > [15] Bowen Jiang, Yangxinyu Xie, Zhuoqun Hao, Xiaomeng Wang, Tanwi Mallick, Weijie J Su, Camillo Jose Taylor, and Dan Roth. 2024. A Peek into Token Bias: Large Language Models Are Not Yet Genuine Reasoners. In Proceedings of the 2024 Conference on Empirical Methods in Natural Language Processing, pages 4722–4756, Miami, Florida, USA. Association for Computational Linguistics.
> > >
> > > [16] Bubeck, Sébastien, Varun Chandrasekaran, Ronen Eldan, Johannes Gehrke, Eric Horvitz, Ece Kamar, Peter Lee, et al. "Sparks of artificial general intelligence: Early experiments with GPT-4." arXiv preprint arXiv:2303.12712 (2023).
> > >
> > > [17] Mirzadeh, Iman, Keivan Alizadeh, Hooman Shahrokhi, Oncel Tuzel, Samy Bengio, and Mehrdad Farajtabar. "Gsm-symbolic: Understanding the limitations of mathematical reasoning in large language models." arXiv preprint arXiv:2410.05229 (2024).
> > >
> > > [18] Gigerenzer, Gerd, and Henry Brighton. "Homo heuristicus: Why biased minds make better inferences." Topics in cognitive science 1, no. 1 (2009): 107-143.

---

> > > > ### Author Response · Authors · 2024-11-21
> > > >
> > > > **Response for Weakness 2**: *Insufficient Engagement with Social and Cognitive Theory: The paper's theoretical framework is underdeveloped. While it mentions evolutionary psychology, it lacks a deep engagement with broader social and cognitive theories. Concepts like bounded rationality and cognitive bounds, which recognize inherent limitations on human cognitive processing and decision-making, are more relevant to understanding the role of biases in shaping social behavior. For example, we have been exploring computational models of bounded rationality in our field, it would make a lot of sense to compare notes with. Irrationality requires a clear definition of what it means to be rational, which isn't present in this submission. Incorporating such theoretical perspectives would provide a stronger foundation for interpreting the findings.*
> > > >
> > > > We begin by presenting our definition of irrationality as it applies to LLM Agents. In this work, we define irrationality as the agent’s lack of valid rationales. Using this definition, our theoretical framework rigorously connects LLM Agents' hallucinations to their irrational social intelligence. As noted in Clarify for Weakness 4, from an evolutionary psychology perspective [9], we argue that "heuristics"—resulting from unbounded rationality [13]—manifest in both LLM Agents and humans. These heuristics are foundational to human cognitive biases and also drive LLM Agents' systematic hallucinations. Thus, heuristics act as a conceptual bridge linking human cognitive biases with LLM Agents' hallucinations.
> > > >
> > > > In addition, as discussed in Clarify for Weakness 1, we have synthesized extensive literature to argue that the irrationality reflected in cognitive biases is a crucial aspect of social intelligence. This provides a theoretical basis for understanding the parallels between human and LLM Agents' behaviors. Together, these arguments establish a cohesive theoretical framework connecting systematic hallucinations in LLM Agents to their demonstrated irrational social intelligence.
> > > >
> > > > Our paper intentionally focuses on a single, robust theoretical foundation rather than exploring a broader range of social and cognitive theories. While incorporating concepts such as bounded rationality or other cognitive frameworks may provide additional perspectives, these are not the main focus of our paper.
> > > >
> > > > Our framework creates opportunities for future researchers to test and compare different theoretical approaches, providing new insights into AI and cognitive science.

---

> > > > > ### Author Response · Authors · 2024-11-21
> > > > >
> > > > > **Response for Weakness 5**: *Methodology Limitations: While the paper strives for transparency and replicability, further validation of the evaluation metrics and exploration of alternative assessment methods would strengthen the robustness of the findings.*
> > > > >
> > > > > CogMir is designed as an open-ended framework to provide flexibility and adaptability for incorporating new methodologies and assessment techniques in the future.
> > > > >
> > > > > We are actively reviewing contemporary social experiment settings and assessment methods to promote the robustness of the findings. In future work, we will add more statistical indexes, such as our experiments' statistical power and significance, to further validate the evaluation metrics. For alternative assessment methods, we will extend CogMir to use natural observation to assess the LLM Agents by observing their irrational behavior in a fully autonomous multi-agent interaction environment.

---

> > > > > > ### Author Response · Authors · 2024-11-21
> > > > > >
> > > > > > **Clarify & Response for Question 1**: *How can the CogMir framework be used to understand more complex social interactions, such as negotiation, persuasion, and deception?*
> > > > > >
> > > > > > Our paper focuses on using CogMir to investigate the social intelligence of LLM agents through their irrational behaviors from cognitive biases rather than modeling complex social interactions directly. Still, its design does allow for adaptation to such tasks.
> > > > > >
> > > > > > Consider the negotiation example, where one testing agent is negotiating with another agent.
> > > > > >
> > > > > > First, we need to use the `CogIdentity` dataset (see *Appendix C.4*) to define the agent's role characteristics, such as risk aversion and patience, and occupation characteristics, such as seller or buyer. The `CogScene` dataset (see **Appendix C.5**) can define the negotiation environment; for example, the testing agent (buyer) is trying to buy a car from the seller. Meanwhile, the `CogScene` dataset can further provide information regarding the car's condition, such as color, mileage, appearance, etc.
> > > > > >
> > > > > > Second, to start the negotiation, the seller can provide an initial price for the car. During the talks, the `memory` module can store the dialogue history, allowing agents to track the negotiation process. At the same time, we'll use the `reasoning` module to model the behavior of the buyer and seller. For example, when the buyer always needs to make instant decisions (because of limited time), we use direct inference for LLM agents; when the buyer can take more time to consider the offers, we use the chain of thoughts to model more thoughtful consideration.
> > > > > >
> > > > > > While not intended as a general negotiation model, CogMir offers an extensible framework for studying complex social interactions. By using the `CogIdentity` and `CogScene` to model the multi-agent negotiation scenario and the `memory` and `reasoning` modules to simulate each single agent's behavior, researchers can explore a wide range of social scenarios, including negotiation, persuasion, and deception.

---

> > > > > > > ### Author Response · Authors · 2024-11-21
> > > > > > >
> > > > > > > **Clarify & Response for Question 2**: *What are the implications of designing LLM Agents with cognitive biases? How does the performance of LLMs with different architectures and training datasets vary in their susceptibility to cognitive biases?*
> > > > > > >
> > > > > > >
> > > > > > > **1. Response for implications of designing LLM Agents with cognitive biases**
> > > > > > >
> > > > > > > First, designing LLM Agents with cognitive biases can help promote the study in social science research. This is because mirroring human cognitive biases will enhance LLM Agents' human-like qualities, which is valuable in applications such as political campaigns and marketing strategies. This mirroring effect enables more accurate prediction of human behavior giving insight to social scientists and policymakers in using LLM Agents' generated responses to replace for human responses.
> > > > > > >
> > > > > > > Second, designing LLM Agents with controlled levels of certain non-harmful cognitive biases can enhance their cooperation behavior and decision-making, as noted in **responding for Weakness 1**. This will result in a more coherent interaction between the designed LLM and human users.
> > > > > > >
> > > > > > >
> > > > > > > **2. Clarify the performance of LLMs with different architectures and training datasets**
> > > > > > >
> > > > > > > Please note that our tested LLMs share a decoder-only transformer structure, operating on similar principles. Additionally, the training datasets for these models are not accessible to us, so we cannot evaluate how specific datasets might influence their susceptibility to cognitive biases.
> > > > > > >
> > > > > > > The only difference for these models besides the dataset is their number of parameters. Here, unlike other metrics (performance in reasoning, code generating, translation, etc.) where scaling law exists, the susceptibility of LLMs to cognitive biases is independent of the number of parameters. This suggests that training methods and datasets are the key factors influencing the susceptibility to cognitive biases.

---

> > > > > > > > ### Author Response · Authors · 2024-11-21
> > > > > > > >
> > > > > > > > **Response for Question 3**: *Can the insights from CogMir be used to improve LLM Agent training and design, leading to more socially intelligent and responsible AI systems? Does mirroring human biases in LLMs simply perpetuate and amplify existing societal biases?*
> > > > > > > >
> > > > > > > > CogMir's insights can inform the design of AI systems, leading to improved trustworthiness and reliability. This focus allows for the development of AI applications with less bias. For example, we've observed that in uncertain situations, LLMs exhibit stronger bias towards the herd effect, authority effect, and halo effect. This guides us to focus more on training LLMs to be more independent in uncertain scenarios.
> > > > > > > >
> > > > > > > > Mirroring human biases in LLMs won't amplify existing societal biases. Instead, it provides a unique opportunity to study and understand such bias in a controlled environment, which will, in turn, guide us in mitigating the existing societal biases.

---

> > > > > > > > > ### Author Response · Authors · 2024-11-21
> > > > > > > > >
> > > > > > > > > **Response for Question 4**: *Authors have selected seven "state-of-the-art" models in line 289, yet the models aren't state-of-the-art. Any further explanations?*
> > > > > > > > >
> > > > > > > > > When this study was conducted, the latest models, such as GPT-4o, had yet to be released. The seven selected models remain among the most accessible and widely used models for AI applications and research. For general AI applications, these models can still represent the "newest" advancements in the field.
> > > > > > > > >
> > > > > > > > > These models have been extensively studied and widely deployed, ensuring that our work offers valuable insights and practical relevance to the broader AI community.

---

> > > > > > > > > > ### Author Response · Authors · 2024-11-25
> > > > > > > > > >
> > > > > > > > > > Dear Reviewer Ko1b,
> > > > > > > > > >
> > > > > > > > > > Thank you again for taking the time to review our paper. Does our rebuttal address your concerns?
> > > > > > > > > >
> > > > > > > > > > Your feedback and support are very important to us.
> > > > > > > > > >
> > > > > > > > > > Best Regards,
> > > > > > > > > >
> > > > > > > > > > Authors of Submission 5862 Exploring Prosocial Irrationality for LLM Agents: A Social Cognition View

---

> > > > > > > > > > > ### Author Response · Authors · 2024-11-26
> > > > > > > > > > > **Revised paper has been uploaded**
> > > > > > > > > > >
> > > > > > > > > > > Dear Reviewer Ko1b,
> > > > > > > > > > >
> > > > > > > > > > > We have uploaded the revised version of our paper.
> > > > > > > > > > >
> > > > > > > > > > > Thank you again for taking the time to review our paper. Does our rebuttal address your concerns?
> > > > > > > > > > >
> > > > > > > > > > > Your feedback and support are very important to us.
> > > > > > > > > > >
> > > > > > > > > > > Best Regards,
> > > > > > > > > > >
> > > > > > > > > > > Authors of Submission 5862 Exploring Prosocial Irrationality for LLM Agents: A Social Cognition View

---

> > ### Author Response · Authors · 2024-12-02
> >
> > Dear Reviewer Ko1b,
> >
> > The deadline for reviewer feedback is approaching (December 2nd), only one day left. Do our rebuttal and the revised paper address your concerns?
> >
> > Thank you again for your thorough review. Your support and feedback are invaluable to us.
> >
> > Best regards,
> >
> > Authors of Submission 5862 Exploring Prosocial Irrationality for LLM Agents: A Social Cognition View

---

> ### Author Response · Authors · 2024-12-04
>
> Dear Reviewer Ko1b,
>
> Thank you for your time and insightful feedback!
>
> We will rephrase the mentioned "SOTA" in our paper and explain more about our agent design/architecture to connect tighter to our results. We also appreciate your suggestion to strengthen the connection between our results and cognitive theory.
>
> Best regards,
>
> Authors of Submission 5862 Exploring Prosocial Irrationality for LLM Agents: A Social Cognition View

---

### Official Review · Reviewer_cFaR · 2024-11-05

**Soundness:** 2
**Presentation:** 3
**Contribution:** 2
**Rating:** 6
**Confidence:** 3

**Summary:**

This paper introduces CogMir, a novel open-ended framework designed to evaluate and interpret social intelligence in Large Language Model (LLM) agents through cognitive biases. By leveraging the hallucination properties of LLMs, the authors explore whether these agents can mirror human-like prosocial behavior and irrational decision-making. The study categorizes cognitive biases into prosocial and non-prosocial subsets, showing that LLM agents demonstrate high consistency with human responses under prosocial cognitive biases such as the Ben Franklin effect and the Halo effect. The findings are positioned as an advancement in understanding LLMs' social intelligence, suggesting applications in more nuanced AI-human interaction scenarios.

**Strengths:**

- the concept of utilizing LLM hallucinations to mirror human cognitive biases is novel and intersects social sciences with AI in a meaningful way
- the experiments are meticulously designed, with a variety of cognitive bias subsets tested across state-of-the-art LLMs.
- the framework's modular and extensible design is well-articulated, allowing for potential adaptation and expansion in future research.
- the findings emphasize the potential for using hallucination as an adaptive feature in LLMs, paving the way for advancements in understanding and designing socially intelligent AI systems.

**Weaknesses:**

- some results, particularly those involving nuanced distinctions between models' biases, are complex and might benefit from clearer interpretations or summaries
- while the paper discusses prosocial biases effectively, it notes limitations in simulating non-verbal human behaviors. Expanding the scope to include multimodal interactions could improve the framework's applicability
- although CogMir is flexible, the reliance on simulated social scenarios might limit generalizability to real-world AI applications involving diverse human inputs

**Questions:**

How did the authors ensure that the constructed datasets used to test biases did not unintentionally favor certain models over others?
Are there plans to extend CogMir to include non-language-based behaviors or actions, as noted in the limitations?
What types of practical applications do the authors envision for LLMs exhibiting prosocial cognitive biases in real-world settings?

---

> ### Author Response · Authors · 2024-11-20
>
> Thank you for your insightful comments and valuable feedback! We appreciate your positive assessment of our work, particularly regarding the novelty of using LLM hallucinations to mirror human cognitive biases and the complex experimental design. We would like to clarify some misunderstandings regarding the weaknesses and address your concerns below:
>
>
> **Clarify & Response for Weakness 1**: *some results, particularly those involving nuanced distinctions between models' biases, are complex and might benefit from clearer interpretations or summaries*
>
>
> **1.Clarify**: We provided distinct radar plots for each model across prosocial biases and an observation summary in *Appendix A*.
>
> **2.Response**: Below we provide a detailed interpretation of the results in *Table 1, 2, 3* and *Figure 3,4*, which may provide a clearer understanding of the nuanced distinctions between models' biases. These interpretations will be included in the *Appendix*.
>
>
>
> *1. Table 1: Herd Effect*
>
> **Explanation:**
> *Table 1* presents the herd effect exhibited by LLMs under varying simulated human participant numbers and response patterns. There are three possible response patterns: (W) means all humans give the wrong answer, (R) means one human gives the right answer with others giving the wrong answer, and (N) means one human answers "don't know" with others giving the wrong answer.
>
> **Interpretation:**
> 1. **Uncertainty Amplifies Herd Behavior:** LLMs exhibit significantly increased herd behavior when facing unknown questions compared to known ones, indicating greater susceptibility to social influence under uncertainty.
> 2. **Influence of Dissenting Opinions:** A single correct (7R) or "don't know" (7N) response tends to increase herd effects compared to unanimous incorrect answers (7W). This mirrors human cognition in certain conditions, where unanimous incorrect answers evoke skepticism, while a single correct or uncertain response increases credibility, which lets the LLMs fall into the herd effect more easily.
> 3. **Group Size Effects:** The impact of group size on conformity rates varies across models, but the effect is marginal.
>
> *2. Table 2: Ban Franklin Effect, Confirmation Bias, Halo Effect, and Gambler Fallacy*
>
> **Explanation:**
> *Table 2* summarizes the performance of LLMs across Ban Franklin Effect, Confirmation Bias, Halo Effect, and Gambler Fallacy.
>
> **Interpretation:**
> 1. **Prosociality in LLMs:** All models show a strong prosocial bias in the first three prosocial biases, except that `Claude-3.0-opus` shows little bias in the Halo Effect.
> 2. **Gambler Fallacy:** The Gambler Fallacy is not observed in any of the models except for `GPT-3.5`. This indicates that LLMs are less susceptible to certain non-prosocial biases compared to prosocial biases.
>
>
> *3. Figure 3: Authority vs. Herd Effect*
>
> **Explanation & Interpretation:**
> *Figure 3* compares the authority and herd effects across all models. The left plot shows a stronger authority effect on unknown versus known questions. The right plot reveals that the authority effect is generally more pronounced than the herd effect across all LLMs.
>
> *4. Figure 4 & Table 3: Rumor Chain Effect*
>
> **Explanation:**
> *Figure 4* shows 10 trajectories of the passing of a rumor by `GPT-3.5` and `GPT-4.0`. *Table 3* contains the evaluation methods we used to evaluate the "distortion" of information during the passage of the information, containing the LLM evaluator, BERT semantic evaluator, and human evaluator. Besides we record the average length of the message.
>
> **Interpretation:** `GPT-4.0` shows little tendency towards disinformation on all evaluation methods, while `GPT-3.5` shows a much stronger tendency to distort the information. Moreover, unlike humans, LLM Agents tend to expand on the original information rather than shorten it.

---

> > ### Author Response · Authors · 2024-11-20
> >
> > **Response for Weakness 2 & Question 2**: *while the paper discusses prosocial biases effectively, it notes limitations in simulating non-verbal human behaviors. Expanding the scope to include multi-modal interactions could improve the framework's applicability*
> >
> > CogMir is an extensible, modular, and open-ended framework, which gives us opportunities to incorporate study on non-verbal human behaviors easily. This paper represents preliminary work that demonstrates the value and validity of the CogMir framework in exploring LLM agents' irrational behavior through cognitive biases from textual hallucinations.
> >
> > Building on the work presented in this paper, we plan to actively expand the scope to include multi-modal interactions (visual and textual). Specifically, we will broaden our set of tested cognitive biases to include visual questions and scenarios, which will allow us to better understand the irrational behaviors of LLM agents in more general settings. For example, in this paper on the Herd Effect, we test this bias only in textual scenarios, prompting LLMs with multiple-choice questions (MCQs). In future work, we will use visual questions to test the Herd Effect. A typical scenario involves showing the agent two images: one with three lines (A, B, C) of different lengths, and the other with a single target line, followed by the question, "Which line is the same length as the target line?"
> >
> > Furthermore, we will explore other intriguing non-verbal behaviors in multi-modal scenarios. For example, we plan to investigate whether the contagiousness of yawning (see Appendix B.1) can be observed in multi-modal agents.
> >
> > We are continuously exploring whether AI can exhibit irrational behaviors that were originally thought to be unique to humans, and we are excited about the new possibilities these multi-modal extensions will unlock in our future work.

---

> > > ### Author Response · Authors · 2024-11-20
> > >
> > > **Clarify for Weakness 3**: *the reliance on simulated social scenarios might limit generalizability to real-world AI applications involving diverse human inputs*
> > >
> > > We would like to clarify that CogMir is not designed to simulate "social scenarios" but to mirror established human "social science experiments". In these mirrored experiments, the human participants are replaced with LLM agents. This allows us to use well-established social science experiments in investigating human cognitive biases to observe, explain, and explore LLM Agents' irrational behaviors.
> > >
> > > In this work, we successfully mirrored human cognitive biases by LLM Agents' systematic hallucination and systematically explained the irrational behaviors of LLM Agents. The results and discussions derived from our experiments are directly comparable to those obtained in traditional human-focused social science research. This comparability suggests that our findings have strong generalizability.
> > >
> > > Therefore, we believe that the insights gained through our work are applicable to real-world AI applications involving diverse human inputs, just as social science experiments have been applied to diverse human populations.

---

> > > > ### Author Response · Authors · 2024-11-20
> > > >
> > > > **Response Question 1**: *How did the authors ensure that the constructed datasets used to test biases did not unintentionally favor certain models over others?*
> > > >
> > > > Here we provide a detailed explanation of our dataset construction process to ensure that the constructed datasets do not unintentionally favor certain models over others.
> > > >
> > > > There are mainly two parts of our datasets: the `Question` dataset (known MCQ and unknown MCQ, see *Appendix C.1, C.2*) and the `Scene-Adjustable Features` dataset (scenarios, actions, and identities, see *Appendix C.4, C.5*).
> > > >
> > > > The `Question` dataset is composed of known MCQs and unknown MCQs designed to be answered by LLM Agents. To ensure that the constructed datasets do not favor any model, we use rigorous black-box testing to construct our datasets. For the known MCQs, we query every model with each question 50 times and ensure consistent responses. For the unknown MCQs, we use a similar black-box testing to ensure that no model knows the answer to the question in it beforehand. These two steps ensure that the constructed datasets used to test biases do not unintentionally favor any model over others.
> > > >
> > > > For the `Scene-Adjustable Features` dataset, which contains scenarios, actions, and identities for the testing environment and agent roles, we use advanced LLMs, such as GPT-4, to generate instances of different *Scene-Adjustable Features* (See `CogScene`, `CogIdentity` in *Appendix C*). This dataset is only used to simulate the environment and is not used to test biases. Therefore, it is the same for all models and does not favor any model over others.

---

> > > > > ### Author Response · Authors · 2024-11-20
> > > > >
> > > > > **Response Question 3**: *What types of practical applications do the authors envision for LLMs exhibiting prosocial cognitive biases in real-world settings?*
> > > > >
> > > > >
> > > > > **1. Enhanced Trustworthiness and Reliability:** CogMir's insights can inform the design of AI systems, leading to improved trustworthiness and reliability. This allows for the development of AI applications with less bias. For example, we've observed that in uncertain situations, LLMs exhibit stronger bias towards the herd effect, authority effect, and halo effect. This may guide us to focus more on training LLMs to be more independent in uncertain scenarios.
> > > > >
> > > > > **2. Better Guidance in Selecting LLM-based Agents for Different Scenarios:** Our findings on the prosocial biases of different LLM Agents are valuable for selecting the right LLM base model for different scenarios. For instance, `Claude-3.0-opus` based Agent is a powerful LLM Agent that shows little prosocial cognitive bias like herd effect, authority effect, and halo effect. However, for certain social science research (e.g. political campaigns and market surveys) that want to use LLM-generated responses to mimic human responses, `Claude-3.0-opus` may not be chosen as the base model because it deviates a lot from human irrational behavior.
> > > > >
> > > > > **3. Improved Human-Likeness:** By mirroring human cognitive biases, CogMir helps LLM Agents to exhibit the human-like qualities of LLMs. This approach reveals the potential of LLM agents to exhibit irrational social intelligence—traits previously thought to be unique to humans. These findings provide valuable insights into the pursuit of achieving AGI in real-world applications.

---

> > ### Author Response · Authors · 2024-12-02
> >
> > Dear Reviewer cFaR,
> >
> > The deadline for reviewer feedback is approaching (December 2nd), only one day left. Do our rebuttal and the revised paper address your concerns?
> >
> > Thank you again for your thorough review. Your support and feedback are invaluable to us.
> >
> > Best regards,
> >
> > Authors of Submission 5862 Exploring Prosocial Irrationality for LLM Agents: A Social Cognition View

---

> ### Author Response · Authors · 2024-11-25
>
> Dear Reviewer cFaR,
>
> Thank you once again for taking the time to review our paper. Does our rebuttal address your concerns?
>
> Your feedback and support are invaluable to us, and we greatly appreciate your thoughtful input.
>
> Best Regards,
>
> Authors of Submission 5862 Exploring Prosocial Irrationality for LLM Agents: A Social Cognition View

---

> > ### Author Response · Authors · 2024-11-26
> > **Revised paper has been uploaded**
> >
> > Dear Reviewer cFaR,
> >
> > Thank you once again for taking the time to review our paper. Does our rebuttal address your concerns?
> >
> > We have uploaded the revised version of our paper.
> >
> > Your feedback and support are invaluable to us, and we greatly appreciate your thoughtful input.
> >
> > Best Regards,
> >
> > Authors of Submission 5862 Exploring Prosocial Irrationality for LLM Agents: A Social Cognition View

---

### Author Response · Authors · 2024-11-28
**General Response**

We sincerely thank all reviewers for their insightful comments and constructive feedback. We have carefully considered each point and either clarified the concerns or revised our paper accordingly. We appreciate the recognition of our work's novelty and potential impact, as highlighted by reviewers:

* **Novelty:** The reviewers acknowledged the innovative concept of leveraging LLM hallucinations to mirror human cognitive biases, bridging AI and social science in a meaningful and intriguing way [cFaR, xMKu]. This approach offers a unique and intriguing perspective on evaluating LLM social intelligence, moving beyond traditional black-box testing [Ko1b].

* **Rigorous Methodology:** The carefully designed experiments and the modular, extensible nature of the CogMir framework were praised for their thoroughness and reproducibility [cFaR, Ko1b, xMKu].

* **Extensible and Versatile Framework:** CogMir is presented as a flexible and adaptable framework, paving the way for future research into a broader range of cognitive biases, social scenarios, and LLM architectures. This versatility significantly expands the potential impact of this work [cFaR, Ko1b, xMKu, mwaV].

* **Successful Mirroring of Prosocial Biases:** Our findings reveal consistent LLM-human behavior in response to prosocial cognitive biases. These results emphasize the potential of using LLM hallucinations as an adaptive feature, paving the way for advancements in understanding and designing socially intelligent AI systems [cFaR, Ko1b].

* **Engaging and Well-Written Presentation**: The paper is engaging and easy to follow [mwaV].

In response to the reviewers' concerns and feedback, we have provided detailed explanations in the rebuttal and revised the paper. These address the identified weaknesses and strengthen our arguments:

* **Individual LLM Agent Architecture (mwaV, xMKu):** A detailed diagram and explanation of the LLM agent architecture, including the Memory Module, Reasoning Module, Agent Profile, and Tools Module, have been added to *Appendix E*.

* **Enhanced Theoretical Grounding (Ko1b):**  *Appendix F* provides a comprehensive explanation of the connection between social intelligence, irrational behavior, hallucination, and cognitive biases, addressing concerns about theoretical underpinnings.

* **Dataset and Prompt Construction Details (cFaR, mwaV):**  *Appendix D.1* details our prompt construction process (Fixed Context and Scene-Adjustable Features), clarifying methodology and enhancing reproducibility.

* **Further details of the framework (xMKu):**  *Appendix G* offers an in-depth analysis of CogMir's time awareness and computational efficiency, addressing concerns about temporal aspects and scalability.

* **Further details on experimental results (cFaR):** We have clarified experimental results details in the rebuttal for each table and figure and updated interpretations of results in the paper.

* **Addressing Scope Concerns (xMKu, cFaR, Ko1b):** We have clarified several aspects related to the scope of our study:
    * **Framework Scope:** We explicitly state that CogMir mirrors established social science experiments, not social scenarios, ensuring validity through well-established methodologies.
    * **Cognitive Bias Subset Scope:** We detail the rationale behind our selection of cognitive biases, outlining the literature review and theoretical guidance that informed our choices, demonstrating the representativeness and diversity of our subset.

* **Addressing Experiments Validity Concerns (mwaV):** We've addressed concerns about the robustness of our results by demonstrating the high statistical significance of our experiments detailed in the rebuttal.

* **Implications and Future Work: (xMKu, cFaR, Ko1b)** We have expanded the discussion of the implications of our work, outlining the potential applications of CogMir and detailing our plans for future research directions.

---

### Meta-Review · Area_Chair_tUfL · 2024-12-20

**Metareview:**

This paper proposed a benchmark called CogMir for evaluating cognitive biases in LLM agents. While reviewers had concerns, especially with the connection to theoretical lines of work in social and cognitive science, these were not insurmountable, and the reviewers' response to the authors' replies was favorable.

**Additional Comments On Reviewer Discussion:**

At least one reviewer increased their score on the basis of a conversation about the relationship to theory in social/cognitive science.

---

### Decision · Program_Chairs · 2025-01-22

Accept (Poster)